# Model of Early Diagenesis in the Upper Sediment with Adaptable complexity – MEDUSA (v. 2): a time-dependent biogeochemical sediment module for Earth System Models, process analysis and teaching

Guy Munhoven

Dépt. d'Astrophysique, Géophysique et Océanographie, Université de Liège, B–4000 Liège, Belgium

**Correspondence:** Guy Munhoven
(Guy.Munhoven@uliege.be)

**Abstract.** MEDUSA is a time-dependent one-dimensional numerical model of coupled early diagenetic processes in the surface sea-floor sediment. In the vertical, the sediment is subdivided into two different zones. Solids (biogenic, mineral, etc.) raining down from the surface of the ocean are collected by the reactive mixed layer at the top. This is where chemical reactions take place. Solids are transported by bioturbation and advection, solutes by diffusion and bioirrigation. The classical coupled time-dependent early diagenesis equations (advection-diffusion reaction equations) are used to describe the evolutions of the solid and solute components here. Solids that get transported deeper than the bottom boundary of the reactive mixed layer enter the second zone underneath, where reactions and mixing are neglected. Gradually as solid material gets transferred here from the overlying reactive layer, it is buried and preserved in a stack of layers that make up a synthetic sediment core.

MEDUSA has been extensively modified since its first release from 2007. The composition of the two phases, the processes (chemical reactions) and chemical equilibria between solutes are not fixed any more, but get assembled from a set of XML based description files that are processed by a code generator to produce the required Fortran code. 1D, 2D and 2D×2D interfaces have been introduced to facilitate the coupling to common grid configurations and material compositions used in biogeochemical models. MEDUSA can also be run in parallel computing environments using the Message Passing Interface (MPI).

## 1 Introduction

### 1.1 Ocean-sediment exchange schemes: an overview

Ocean biogeochemical cycle models call upon a variety of schemes of different complexity levels to represent ocean-sediment exchange fluxes. These can be classified into four major categories (Hülse et al., 2017): (1) reflective boundary conditions; (2) semi-reflective or conservative; (3) vertically-integrated dynamic models; (4) vertically resolved diagenetic models. These categories are similar but not identical to the levels in the classification of Soetaert et al. (2000): categories 3 and 4 respectively correspond to their level 3 and 4 descriptions; category 1 fits their level 2 description, while category 2 generalises the latter.

Reflective boundary conditions are the simplest of these: material reaching the model sea-floor (i.e., the deepest layer in the model water column) is unconditionally remineralised (oxidized, dissolved) there. Global mass conservation is obviously guaranteed with this approach, but the approach may be unrealistic in some places: calcite gets dissolved even if the sea-floor bathes in waters that are strongly supersaturated with respect to calcite or organic matter oxidized even if oxygen levels are low. This unrealistic behaviour can to some extent be alleviated with the semi-reflective or conservative scheme. Here, only part of the remineralisation products (nutrients, dissolved inorganic carbon, silica, etc.) of the solids reaching the sea-floor are returned to the bottom water; the remainder is returned to the surface, mimicking riverine input, and once again allowing for global mass conservation. The fraction remineralised can be made to vary in space and time and can also be different for different materials. Carbonate fractions remineralised can, e.g., be linked to the degree of saturation with respect to one carbonate mineral or another, and organic carbon fractions remineralised to the degree of oxygenation of the bottom waters. Both schemes are attractive because of their convenient computational efficiencies. They do, however, not allow to take into account the complexities of the actual remineralisation pathways of the various biogenic components in the surface sediments, nor can they represent the temporary storage of such material in the surface sediment and delayed return of nutrients, dissolved inorganic carbon or silica to the ocean bottom waters.

The vertically-integrated dynamic category 3 encompasses ocean-sediment exchange schemes that explicitly include a single-box representation of the surface sediment. Mass balances of some, if not all, constituents of this single-layer sediment can be traced. Although termed "vertically-integrated" not all of the schemes that fall into this category can be traced back to some actual vertically resolved model that was vertically integrated.

Vertically resolved diagenetic models finally represent the most mechanistically oriented alternative to represent ocean-sediment exchange fluxes. Such models can take into account the complex interplay between various diagenetic processes (organic matter remineralisation, mineral dissolution or precipitation) and transport pathways (advection, bioturbation, solute diffusion in porewater, bioirrigation, etc.). They solve a set of coupled standard early diagenesis equations (Boudreau, 1997) for solid and dissolved component concentrations, generally in combination with law of mass action equations for chemical equilibria (e.g., for the carbonate system).

Meta-model approaches (Sigman et al., 1998; Dunne et al., 2007; Ridgwell, 2007; Capet et al., 2016), i.e., parametric representations or emulators of comprehensive models, may either fit into the second or the third categories, depending on their design. Such emulators generally come as empirical parametric functions, typically derived by fitting selected model outcomes (such as diffusive return fluxes) from large sets of simulation experiments carried out with varying boundary conditions to some expert-chosen empirical parametric functions (Dunne et al., 2007). Another promising venue is the analysis of complex models with approaches based upon system identification theory (see Crucifix (2012) for an introduction to these methods to the emulation of complete ESMs and Ermakov et al. (2013) for a pilot application to the coupled ocean carbon cycle-sediment model MBM-MEDUSA (Munhoven, 2007)).

The actually required complexity of an adopted ocean-sediment exchange scheme depends of course on the application made. For short-term experiments (say a few decades to a few centuries) with high-resolution biogeochemical models carefully calibrated semi-reflective/conservative schemes are generally the only viable, but nevertheless perfectly acceptable option.

For long-term applications (simulation experiments exceeding several thousands of years, i.e., several ocean mixing cycles), vertically integrated or resolved schemes are required for realistic model responses to changing boundary conditions and forcings.

The surface sedimentary mixed-layer, where most of the processing of the deposited biogeochemically relevant material takes place only extends down to about 5 to 15 cm on global average ($9.8 \pm 4.5$ cm according to Boudreau (1994)). As a result of the diagenetic processes in action there, strong concentration gradients are generated and sustained: the amplitude of the concentration differences observed over this depth interval may be comparable to those observed in the more than four orders of magnitude thicker overlying water column (3750 m on average) – see, e.g., the oxygen and pH profile data in Sect. 3.3 below.

A realistic explicit representation of the surface sedimentary environment thus requires a vertical resolution of the same order of magnitude in terms of vertical layers or grid points than the complete water column above it. Typical vertically resolved early diagenesis models present vertical resolutions of the order of ten to twenty layers (see Table 1). For comparison the water column in GENIE-1, which includes SEDGEM, (Ridgwell, 2007; Ridgwell and Hargreaves, 2007) has eight vertical layers, HAMOCC-2S (Heinze et al., 2009) has ten and the more recent HAMOCC 5.2 (Ilyina et al., 2013a) has forty layers.

Accordingly, sea-floor sediment modules (category 3 and 4 schemes) are not yet commonplace in global ocean biogeochemical models. Only four out of fifteen Earth System Models of Intermediate Complexity (EMICs) participating in the EMIC AR5 Intercomparison Project (Eby et al., 2013) are reported to have a sediment module included: Bern3D (Tschumi et al., 2011), DCESS-ESM (Shaffer et al., 2008), GENIE (Ridgwell and Hargreaves, 2007) and UVic 2.9 (Eby et al., 2009); one further participating EMIC, CLIMBER 2, is also routinely used with a sedimentary module included (e.g., Brovkin et al., 2012). Advanced

high-resolution models generally call upon category 1 or 2 schemes for their sea-floor boundary condition, although there are exceptions. The HAMOCC (HAmburg Model of the Oceanic Carbon Cycle) family of models, whose origins reach back to Maier-Reimer (1984), actually has a long-standing history of explicitly taking sedimentary processes into account. HAMOCC 2 (Heinze et al., 1991) was the first one to get a fully coupled sediment module (Archer and Maier-Reimer, 1994), the oxic-only version of the calcite dissolution model of Archer (1991). Later, it received a purposely developed sediment module (Heinze

et al., 1999; Heinze and Maier-Reimer, 1999). Archer et al. (2000) use HAMOCC 2 coupled to the much more complete diagenetic model MUDS (Archer et al., 2002), which considers the sequence of oxic, suboxic (via $NO_3^-$, FeOOH and $MnO_2$ reduction) and anoxic (via $SO_4^{2-}$ reduction) organic matter remineralisation pathways. Later developments of HAMOCC also included suboxic remineralisation pathways of organic matter in the standard sediment model of HAMOCC: in HAMOCC 5.1 (Maier-Reimer et al., 2005) denitrification was added, in HAMOCC 5.2 (Ilyina et al., 2013b) sulfate reduction. Gehlen et al.

(2006) have coupled a slightly extended version of the sediment model of Heinze et al. (1999) to PISCES, the biogeochemical component of NEMO, with nitrate reduction and denitrification as an additional remineralisation pathway for organic matter. The Gehlen et al. (2006) model was later also introduced as the sediment component into Bern3D (Tschumi et al., 2011).

Box models and box-diffusion models of the ocean carbon cycle have an even longer-standing history of including ocean-sediment exchange schemes. Especially box models have been for a long time the only types of models that could be used to

carry out analyses on time scales of several thousands to tens or hundreds of thousands of years. Hoffert et al. (1981) outline the fundamentals of a simple ocean-sediment exchange scheme for their box-diffusion carbon cycle model, but without actually

using it in the end and so Keir and Berger (1983) appear to have been the first to couple a vertically-integrated sediment model to a two-box representation of the ocean-atmosphere carbon cycle for their study of glacial-interglacial $CO_2$ variations. The theoretical foundations of that scheme had been presented in Keir (1982) (see also Munhoven (1997) for a variant and additional

details). In that scheme, the surface sediment was assumed to be well-mixed, with clay and calcite as the only solid components, and with $CO_3^{2-}$ as the only modelled solute in the porewater. Adopting furthermore a calcite dissolution rate proportional to $f(1-\Omega)^n$, where $f$ was the calcite fraction in the solid phase, $\Omega$ the degree of supersaturation with respect to calcite and $n$ the dissolution rate order, the steady-state porewater profile of $CO_3^{2-}$ can be calculated. The total dissolution rate can then be set equal to the diffusive flux of $CO_3^{2-}$ at the sediment-water interface (SWI), which is proportional to $\sqrt{f}(1-\Omega_{\mathrm{SWI}})^{\frac{n+1}{2}}$.

This same scheme and variants of it have afterwards been used in a large variety of box and box-diffusion models, with increasingly better geographical resolution as time evolved: Sundquist (1986), with unreported $n$, CYCLOPS (Keir, 1988) with $n = 4.5$, Walker and Opdyke (1995) with $n = 1$, MBM (Munhoven and François, 1996; Munhoven, 1997) with $n = 4.5$. Sigman et al. (1998) reconsidered the CYCLOPS model of Keir (1988) and replaced the purely $CO_3^{2-}$ driven dissolution scheme by a meta-model based upon a multivariate polynomial expression fitted to the calcite dissolution rates obtained with the model of

Martin and Sayles (1996) under various boundary conditions, also capable of taking into account the effect of porewater $CO_2$ derived from the respiration of organic matter on promoting calcite dissolution in the sedimentary mixed-layer. Munhoven (2007) finally replaced the 304 vertically-integrated sediment boxes in MBM by as many vertically resolved and fully coupled MEDUSA-v1 sediment columns (see Sect. 1.2 below for details). The ocean-sediment exchange schemes in all of the three MBM versions furthermore tracked the history of deposition of the sediment solids and could thus consistently take into account the

effect of chemical erosion events.

There are various means to alleviate the computational overburden caused by adding a vertically resolved early diagenesis model to a biogeochemical ocean model. First of all, the number of vertical layers and of chemical constituents or the complexity of the reaction network can be reduced. Most EMICs that include a vertically resolved sediment module appear to follow that pathway (see Tables 1 and 2): UVic 2.9 (Eby et al., 2009) and CLIMBER 2 (Brovkin et al., 2012) both include the

oxic-only model of Archer (1996a), with thirteen vertical layers; DCESS-ESM (Shaffer et al., 2008) includes a hybrid category 3/4 ocean-sediment exchange scheme considering $CO_3^{2-}$, $O_2$ and organic carbon distributions in seven layers, and calcite and clay contents vertically integrated. Shaffer et al. (2008) furthermore use parametrised exponential concentration profile solutions in each layer. Parameter values are then chosen on the basis of concentration and flux continuity considerations at the layer boundaries to assemble the different pieces into a the final concentration profiles. Hülse et al. (2018) adopted a

somewhat similar strategy for OMEN-SED which complements the carbonate preservation scheme SEDGEM in cGENIE (the carbon-centric version of GENIE) with an organic matter preservation scheme. Instead of piecewise analytical concentration profiles as in DCESS, OMEN-SED uses piecewise analytical organic matter reaction rate profiles in the four redox layers and assembles the resulting partial concentration profiles on the basis of similar continuity conditions. For the coupling of OMEN-SED with cGENIE, the overall early diagenetic reaction network was further simplified by neglecting the impact of organic carbon

respiration on carbonate dissolution. One may also reduce the number of spatially distributed sediment columns. This approach was adopted for GEOCLIM *reloaded* (Arndt et al., 2011). The ocean module of GEOCLIM *reloaded* consists of an advective-

diffusive inner ocean, completed by two two-box (surface and deep) ensembles for the polar and epicontinental seas. The inner ocean is divided into several hundreds of $10\,\mathrm{m}$ thick layers. The ocean-sediment exchange scheme, however, consists of only three vertically resolved sediment columns, attached to the polar, the inner and the epicontinental water-columns, respectively. In each of the three sediment columns the complete cascade of organic matter oxidation pathways from aerobic respiration to $NO_3^-$, $Mn(IV)$, $Fe(III)$ and $SO_4^{2-}$ reduction to $CH_4$ formation as well as a series of secondary redox reactions are taken into account. Even with this strongly reduced resolution of the ocean-sediment exchange scheme, the computation impact remains considerable: Arndt et al. (2011) report that one day of CPU allowed for a $1\,\mathrm{Myr}$ simulation without sediments, but only for a $100\,\mathrm{kyr}$ simulation with sediments. Finally, the ocean-sediment exchange scheme and the ocean biogeochemical calculations may be carried out with different time steps (asynchronous coupling). This approach is followed in GENIE (Ridgwell, 2007; Ridgwell and Hargreaves, 2007) and MBM (Munhoven and François, 1996; Munhoven, 1997, 2007).

Sea-floor sediments are not only relevant as "processing units" for biogenic material raining down from the surface euphotic layer, during which some parts get remineralised (i.e., oxidised or dissolved), and the rest gets buried. Burial is, however, at first only temporary. Changes in the overlying boundary conditions (e.g., saturation conditions) may indeed lead to chemical erosion episodes where the surface sedimentary mixed-layer loses material faster than it is replenished by deposition from the water-column above. We are currently at the onset of such an episode: as ocean acidification due to the uptake of anthropogenic $CO_2$ progresses to the deep-ocean, the resulting change in the degree of saturation with respect to carbonate minerals is expected to enhance the dissolution of carbonates in the sea-floor surface sediments at depth so strongly that the dissolution rate will exceed the rate at which carbonate material gets deposited at the sediment water interface (Archer et al., 1998). Previously buried carbonates will then return to the sedimentary mixed layer as a result of the bioturbation activity, which tends to keep the surface mixed layer at a rather stable thickness, which seems to be controlled by the supply of organic matter (Boudreau, 1998). Archer (1996b) estimates that existing carbonates in surface sea-floor sediment can neutralize about $1600\,\mathrm{GtC}$, considerably more than the $\sim 1000\,\mathrm{GtC}$ that may at most be emitted while still limiting global anthropogenic temperature change below $2\,^\circ\mathrm{C}$ (e.g., Zickfeld et al., 2016), but much less than the estimated resources of fossil fuels of $8{,}543 - 13{,}649\,\mathrm{GtC}$ (Bruckner et al., 2014, Table 7.2).

Finally it should not be forgotten that sea-floor sediments represent our most comprehensive source of information about past climate change and it is of course indispensable to understand how early diagenetic processes influence the sedimentary record. It would be desirable to directly compare generated model (synthetic) sedimentary records to the observed records thus opening new possibilities in terms of data assimilation.

## 1.2  MEDUSA: from version 1 to version 2

The first version of the Model of Early Diagenesis in the Upper Sediment,[1] MEDUSA – hereafter MEDUSA-v1 – was described in Munhoven (2007). It is a time-dependent vertically resolved biogeochemical model of the early diagenesis processes in the sea-floor sediment. MEDUSA-v1 included clay, calcite, aragonite and organic matter as solid components, and $CO_2$, $HCO_3^-$,

---

[1]The final 'A' did not have any particular meaning initially, although music lovers amongst early diagenetic modellers will undoubtedly have read it as "A♭ minor."

$CO_3^{2-}$ and $O_2$ as porewater solutes. Besides that configuration, two others (unpublished) had been developed: one which furthermore included opal and dissolved silica and one which also included the $^{13}C$ isotopic signatures of all carbon-bearing components.

Right from the beginning, MEDUSA had been developed as a sediment module for the diverse ocean biogeochemical models used in our research group, ranging from box to three-dimensional models, the latter with diverse grid configurations and also various sets of chemical tracers. Furthermore, our research interests required a model that could be used in studies dealing with time scales ranging from tens of years to hundreds of thousands of years. Accordingly, the model requirements were laid out as follows: (1) the model code should be customizable to accommodate different chemical compositions; (2) the model should offer the possibility to be coupled to strongly different host model grid layouts; (3) it should be possible to run the model with variable time-steps; (4) the model must be able to cope with chemical erosion, i.e., be able to recover previously buried material from deeper layers and to return it to the chemically reactive mixed-layer.

The customization options offered by MEDUSA-v1 had to be selected with pre-processor directives in the code. Extending the capabilities of the model on the basis of that mechanism had become more and more cumbersome and difficult to manage with time. The code was therefore revised in depth and only the parts related to the transport terms in the equations and the equation system solver—the framework system—were kept. The rest of the code was from now on built on purpose for each application with a code configuration and generation tool that would produce and assemble the parts related to the components, processes (reactions) and chemical equilibria required. A code generator was developed to read in the required information, such as chemical and physical properties of components, chemical reactions describing diagenetic processes and chemical equilibria from a series of description files. These description files use a format based upon the eXtensible Markup Language (XML) syntax (W3C, 2008). Organising the information in an XML tree offers attractive flexibility: such a tree can be easily extended for later developments and it is possible to access any particular information wherever it is located in a file. XML thus offers a high degree of compatibility between subsequent versions of the configurator, which can always extract the relevant information as long as the required mark-up tags remain present. Above all, XML files remain mostly human-readable and the possibility to insert comments makes it possible to ensure the traceability of the stored information.

As the complete tool was meant to require only a Fortran compiler to be built, a simple library, called $\mu$XML, for reading and processing basic ASCII encoded XML files in Fortran 95 was developed as a prerequisite.

## 2 Model Description

### 2.1 Vertical partitioning of the sediment column

The complete sediment column is subdivided into three (or four) different vertically stacked parts (called realms), as illustrated on Fig. 1: (1) REACLAY, the top-most part extending downwards from the sediment top at the sediment-water interface and where the chemical reactions are taken into consideration; (2) TRANLAY, the transition layer of changing thickness just underneath, acting as a temporary storage to connect REACLAY to the underlying (3) CORELAY, a stack of sedimentary layers representing the deep sediment, i. e., the sediment core. Additionally an optional diffusive boundary layer (DBL—not to scale

on Fig. 1) acting as a diffusive barrier to the sediment-water exchange of solutes can be included on top of the REACLAY realm. REACLAY includes the bioturbated sedimentary mixed-layer, where most of the reactions relevant for early diagenesis take place (organic matter remineralisation, carbonate dissolution etc.).

 ### 2.1.1 Equations in the DBL and REACLAY realms

In the REACLAY realm, MEDUSA solves the standard time-dependent diagenetic equation (e.g., Berner, 1980; Boudreau, 1997), which can be written for each sediment component $i$ (solid or solute) in generic form as

$$\frac{\partial \hat{C}_i}{\partial t} + \frac{\partial \hat{J}_i}{\partial z} - \hat{S}_i = 0. \tag{1}$$

In this equation, $t$ is time and $z$ depth below the SWI (positive downwards – see Fig. 1). $\hat{C}_i$ denotes the concentration of $i$ in moles for solutes and in kg for solids per unit volume of total sediment (solids plus porewater). $\hat{J}_i$ is the local transport (advection and diffusion), per unit surface area of total sediment. $\hat{S}_i = \hat{R}_i + \hat{r}_i + \hat{Q}_i$ represents the net source-minus-sink balance for constituent $i$ per unit volume of total sediment, where $\hat{R}_i$ is the net reaction rate, equal to the difference between production and destruction (or decay) rates, $\hat{r}_i$ is the net fast reaction rate, that is going to be filtered out of the equations by equilibrium considerations and $\hat{Q}_i$ the non-local transport (considered only for solutes). The total sediment concentrations $\hat{C}_i$ are related to the more directly accessible phase-specific concentrations $C_i^{\text{s}}$ (for solids) and $C_i^{\text{f}}$ (for solutes) by $\hat{C}_i = \varphi^{\text{s}} C_i^{\text{s}}$ and $\hat{C}_i = \varphi^{\text{f}} C_i^{\text{f}}$, respectively. $\varphi^{\text{s}}$ and $\varphi^{\text{f}}$ denote the volume fraction of bulk solids and of porewater in the total sediment, linked to porosity $\varphi$ by $\varphi^{\text{s}}(z) = 1 - \varphi(z)$ and $\varphi^{\text{f}}(z) = \varphi(z)$. The porosity profile $\varphi(z)$ is prescribed but may be different for each column in multi-column set-ups.

In the DBL (if any), only equations for solutes are considered and porosity is set to 1. Solids are supposed to rain through the DBL and directly enter REACLAY at its top surface.

### Chemical Reactions and Equilibria

The set of chemical reactions and equilibria to consider is completely dependent upon the given application, i.e., on the chemical composition of the sediments required, the diagenetic processes to consider (e.g., organic matter remineralisation, possibly following several pathways, carbonate dissolution, . . . ) and the equilibria between components of solute systems (e.g., carbonate, phosphate, borate systems) to take into account. MEDUSA does not include a standard composition and reaction/equilibrium network but must be configured to fit the complexity requirements of a given application: including one or more classes of organic matter (solid or dissolved), one or more types of carbonates, one or more organic matter degradation processes, etc.

The chemical interconversion reactions represented by the $\hat{r}_i$ terms in the source-minus-sink term $\hat{S}_i$ are orders of magnitudes faster than all other reactions. They are supposed to evolve in quasi-equilibrium. The $\hat{r}_i$ are therefore eliminated from the partial differential equation system by considering appropriate linear combinations of selected equations and by including the thermodynamic equilibrium equations in the system of equations. The partial differential equation system is thus converted into a differential algebraic equation (DAE) system. The subroutines required to evaluate the source and sink terms related to

chemical reactions and to convert the complete system to a DAE system are generated by the companion MEDUSA COnfiguration and COde GENeration tool (MEDUSACOCOGEN) described in Sect. 2.4 below.

**Transport**

Solids are transported by advection throughout the sediment column and subject to bioturbation in the surface mixed layer. Bioturbation is represented as a diffusive process. Both inter- and intraphase biodiffusion variants (Boudreau, 1986) are taken into account and can be combined. With interphase biodiffusion the bulk sediment gets mixed by infaunal activity, solids and porewater alike, and porosity gradients are thus affected as well; with intraphase biodiffusion only the solid phase constituents

get mixed:

$$\hat{J}_i = -D_i^{\mathrm{inter}} \frac{\partial \varphi^{\mathrm{s}} C_i^{\mathrm{s}}}{\partial z} - \varphi^{\mathrm{s}} D_i^{\mathrm{intra}} \frac{\partial C_i^{\mathrm{s}}}{\partial z} + \varphi^{\mathrm{s}} w C_i^{\mathrm{s}}.$$

Here, $D_i^{\mathrm{inter}}$ and $D_i^{\mathrm{intra}}$ are respectively the inter- and intraphase biodiffusion coefficients of the solid $i$; $w$ is the solids' advection rate. We suppose that the biodiffusion coefficients within a given sediment column are the same for all solids: $D_i^{\mathrm{inter}}(z) \equiv D^{\mathrm{inter}}(z)$ and $D_i^{\mathrm{intra}}(z) \equiv D^{\mathrm{intra}}(z)$. For convenience, we define $D^{\mathrm{bt}}(z) = D^{\mathrm{inter}}(z) + D^{\mathrm{intra}}(z)$ and $D^{\mathrm{inter}}(z) =$

$\beta(z) D^{\mathrm{bt}}(z)$ where $\beta(z)$ sets the interphase fraction of the biodiffusion process ($\beta = 0$ for the intraphase and $\beta = 1$ for the interphase end-members). After application of the chain rule to the derivative in the interphase diffusion term, $\hat{J}_i$ becomes

$$\hat{J}_i = -\varphi^{\mathrm{s}} D^{\mathrm{bt}} \frac{\partial C_i^{\mathrm{s}}}{\partial z} + \left( \varphi^{\mathrm{s}} w - \beta D^{\mathrm{bt}} \frac{\partial \varphi^{\mathrm{s}}}{\partial z} \right) C_i^{\mathrm{s}}. \tag{2}$$

The advection rate profile $w(z)$ is derived from the depth-integrated solid phase volume conservation equation

$$\varphi^{\mathrm{s}}(z) w(z) - \beta(z) D^{\mathrm{bt}}(z) \frac{\partial \varphi^{\mathrm{s}}}{\partial z} = \sum_{i \in \mathcal{I}^{\mathrm{s}}} \vartheta_i \hat{I}_i^{\mathrm{top}} + \int_{z_T}^{z} \sum_{i \in \mathcal{I}^{\mathrm{s}}} \vartheta_i \hat{R}_i(z') \, dz', \tag{3}$$

where $\mathcal{I}^{\mathrm{s}}$ denotes the inventory of solid components considered in the model configuration, $\vartheta_i$ the partial specific volume of solid $i$ and $\hat{I}_i^{\mathrm{top}}$ its deposition rate per unit surface of total sediment per unit time, entering the surface sediment through the sediment-water interface at the top. We suppose that the densities $\rho_i$ of individual solid components are constant and independent of each other. In this case $\vartheta_i = 1/\rho_i$ and the $\vartheta_i$ are also constant and they commute with the partial derivatives. Equation (3) is obtained by considering the sum of all the solids' evolution equations (1), weighted by the respective partial

specific volumes, together with the static volume conservation equation

$$\sum_{i \in \mathcal{I}^{\mathrm{s}}} \vartheta_i C_i^{\mathrm{s}} = 1. \tag{4}$$

In the current version of MEDUSA porosity profiles are assumed to be at steady state, although this might change in the future. For non steady-state porosity profiles, the right-hand side of Eq. (3) has to be reduced by $\int_{z_T}^{z} \frac{\partial \varphi^{\mathrm{s}}}{\partial t} \, dz'$.

Although the density of each solid constituent is constant, the average density of the solid phase may vary both in space and

time as chemical reactions proceed and modify the sediment composition, and thereby influence the advection rate profiles.

To ensure compatibility with other early diagenesis models which assume that the solid phase has a constant density (both in space and time) and that the effect of chemical reactions on the sediment sediment mass (and volume) is negligible, all the reactive (non inert) solid components can optionally be declared as volumeless.

Solutes are transported by molecular and ionic diffusion in porewaters, by interphase bioturbation, porewater advection and by bioirrigation. The complete expression for the local transport term of a porewater solute $i$ would thus write

$$\hat{J}_i = -\varphi^{\mathrm{f}} \frac{D_i^{\mathrm{sw}}}{\theta^2} \frac{\partial C_i^{\mathrm{f}}}{\partial z} - D_i^{\mathrm{inter}} \frac{\partial \varphi^{\mathrm{f}} C_i^{\mathrm{f}}}{\partial z} + \varphi^{\mathrm{f}} u C_i^{\mathrm{f}},$$

where $D_i^{\mathrm{sw}}$ is the free diffusion coefficient of the solute $i$ in seawater, $\theta^2$ is tortuosity and $u$ is the porewater advection rate. Applying the chain rule to the second term on the right-hand side and collecting similar terms, we get

$$\hat{J}_i = -\varphi^{\mathrm{f}} \left( \frac{D_i^{\mathrm{sw}}}{\theta^2} + \beta D^{\mathrm{bt}} \right) \frac{\partial C_i^{\mathrm{f}}}{\partial z} + \left( \varphi^{\mathrm{f}} u - \beta D^{\mathrm{bt}} \frac{\partial \varphi^{\mathrm{f}}}{\partial z} \right) C_i^{\mathrm{f}}.$$

In the absence of impressed flow, transport by porewater advection is, however, negligible compared to diffusion; biodiffusion coefficients are furthermore an order of magnitude lower than molecular and ionic diffusion coefficients. The expression for the local transport term of a porewater solute $i$ adopted in MEDUSA then reduces to

$$\hat{J}_i = -\varphi^{\mathrm{f}} \frac{D_i^{\mathrm{sw}}}{\theta^2} \frac{\partial C_i^{\mathrm{f}}}{\partial z}. \tag{5}$$

Tortuosity is parametrized as a function of porosity and the diffusion coefficients of individual solutes are calculated as a function of temperature, corrected for pressure and salinity by using the dynamic viscosity.[2]

It should be noticed that in the current version of MEDUSA, neglecting advection and the effect of interphase biodiffusion on solutes contributes to ensure a more precise mass balance. Porewater advection could still be considered for steady-state applications, where $u_{tot} = u - \frac{\beta D^{\mathrm{bt}}}{\varphi^{\mathrm{f}}} \frac{\partial \varphi^{\mathrm{f}}}{\partial z}$ would always be oriented downwards. However, in transient simulation experiments, where $u$ may temporarily be oriented upwards (unburial, chemical erosion), porewaters would flow into the REACLAY realm, requiring the knowledge of solute concentrations below the modelled domain. These latter are, however, currently not tracked.

Bioirrigation provides a non-local transport mode for solutes. In MEDUSA, the source-sink approach (Boudreau, 1984) is used to quantify the effect of bioirrigation:

$$\hat{Q}_i = \alpha \varphi^{\mathrm{f}} (C_i^{\mathrm{oc}} - C_i^{\mathrm{f}}).$$

Here $\alpha$ is the bioirrigation "constant", which may be depth-dependent, and $C_i^{\mathrm{oc}}$ the concentration of solute $i$ in the irrigation channels, set equal to the solute's concentration in the seawater overlying the sediment.

**Boundary conditions**

The differential equation systems describing the evolutions of the compositions of the DBL and REACLAY realms have to be completed by boundary conditions connecting them to the overlying seawater, the underlying TRANLAY and between each

---

[2]Details about these calculations can be found in Sect. 2.4.2 in the technical report "*Early Diagenesis in Sediments. A one-dimensional model formulation*" in the Supplement.

other. Solute concentrations at the top are derived from those in the overlying ocean water ($C_i^{\mathrm{f}}(z_{\mathrm{W}},t) = C_i^{\mathrm{oc}}(t)$ – a Dirichlet boundary condition), prescribed at the top of the DBL if the model set-up includes one and directly at the SWI if not. If a DBL is included in the model setup solute concentrations at the SWI (i.e., at the interface between the DBL and the REACLAY realm) are derived by assuming concentration and flux continuity across that interface (Cauchy boundary condition). Solids reaching the sea-floor are assumed to "rain through" the DBL (if any) and enter the sediment only at the SWI where flux continuity is used as a boundary condition (leading to a Robin boundary condition).

At the bottom of the REACLAY realm, flux continuity is adopted for both solutes and solids. For solutes, this requires the concentration gradient to reduce to zero (Neumann boundary condition) since $u(z) \equiv 0$ is adopted here. For solids, a variety of effective boundary conditions arise – and the types may change in time – depending on whether biodiffusion has vanished there or not and whether the sediment is burying ($w_{\mathrm{B}}^{+} > 0$, where $w_{\mathrm{B}}^{+}$ denotes the advection rate on the outer side of REACLAY's bottom interface) or eroding ($w_{\mathrm{B}}^{+} < 0$) solid material at the bottom of REACLAY.

### 2.1.2 TRANLAY and CORELAY

TRANLAY is a TRANsition LAYer that collects the solids leaving REACLAY through the bottom (see Fig. 1). As soon as its thickness exceeds a given threshold value (by default $1\,\mathrm{cm}$) at the end of a time step, one or more new sediment core layers are formed and subtracted from TRANLAY to be transferred to CORELAY, which is managed as a last-in-first-out stack of sediment layers.

In general, material is only preserved in TRANLAY and CORELAY; it is assumed that no chemical reactions take place there. However, we have to make one exception to this rule: reactions that are part of a radioactive decay chain are still taken into account in these two realms to avoid physically unrealistic results. Radioactive material that would have left REACLAY and be returned there later during a chemical erosion event would contribute to create unrealistically young concentrations in REACLAY if radioactive decay would have been temporarily suspended for a more or less extended time.

## 2.2 Numerical Solution of the Equation System

The complete solution of one sediment column requires the joint solution of three numerical problems: (1) a DAE system in the REACLAY realm, or the combined REACLAY-DBL realms if a DBL is included; (2) a system of ordinary differential equations in the TRANLAY realm; (3) a stack management problem in CORELAY. The three problems are interdependent. If the sediment column is accumulating, the burial flux, i. e., the solids' advection flux across the bottom of REACLAY feeds TRANLAY and in addition new layers for CORELAY are separated from TRANLAY; if the sediment column is eroding, REACLAY is replenished by TRANLAY through its bottom and TRANLAY is replenished by the topmost layers from CORELAY if necessary.

### 2.2.1 REACLAY (and DBL)

The DAE system is solved by using an implicit Euler method for the time dimension and a finite volume method for the spatial dimension. For this purpose, the REACLAY and the DBL realms are partitioned into cells (finite volumes) using a so-called

vertex-centred grid. Each one of these two realms is overlaid by an irregularly spaced grid of points, called nodes: each node is representative of a cell. The cell boundaries, called vertices, are located midway between adjacent nodes. The concentrations of the considered solute and solid components are evaluated at the nodes, the fluxes between cells at the vertices. The actual grid-point distribution is obtained by a continuously differentiable mapping of a regular grid in order to make sure that the discrete representation of the equation system is consistent and has the same discretisation order than it would have on a regular grid (Hundsdorfer and Verwer, 2003).

The bottom point of the grid that covers REACLAY is always a node. The nature of the topmost point depends on whether a DBL is included or not: if no DBL is included, the topmost interface of REACLAY—the sediment-water interface, SWI—is located at a grid node; if a DBL is included, the interface at the top of REACLAY is mapped onto a grid vertex, defined by a virtual node located above the top of REACLAY and the topmost interior node of REACLAY. Similarly, the bottom of the DBL is located at a vertex of the DBL grid, defined by a virtual node located below the bottom of the DBL and the lowest node inside the DBL (which is most often the top of the DBL). Detailed information about the grid generation can be found in the "MEDUSA *Technical Reference*" in the Supplement.

In multi-column setups—this would be the most common usage for coupling to biogeochemical cycle models—every sediment column in MEDUSA must have the same number of grid-points, but the spacing and extent of each of these may be different.

**Discrete equations**

The discrete version of the evolution equation for a component $i$ in a cell $j$ represented by the node situated at $z_j$ writes

$$\frac{(\hat{C}_i)_j^n - (\hat{C}_i)_j^{n-1}}{\Delta t_n} + \frac{(\hat{J}_i)_{j+\frac{1}{2}}^n - (\hat{J}_i)_{j-\frac{1}{2}}^n}{h_j} - (\hat{S}_i)_j^n = 0, \tag{6}$$

where $\Delta t = t_n - t_{n-1}$ is the implicit time step and $h_j$ is the distance between the vertices $z_{j-\frac{1}{2}} = \frac{1}{2}(z_j + z_{j-1})$ and $z_{j+\frac{1}{2}} = \frac{1}{2}(z_{j+1} + z_j)$ that delimit the cell $j$. Equation (6) is slightly modified at the bottom-most node where it relates to a half-cell only and one may chose to formally express the mass-balance equations for that half-cell at the representative node (which is actually the bottom node of the grid, resp.), or at some intermediate point between that node and the delimiting vertex. In the absence of a DBL, a similar procedure is adopted at the top-most node.

The numerical schemes adopted in MEDUSA have been selected with the physical meaningfulness of the results in mind. Accordingly, positiveness of the calculated concentration evolutions was deemed indispensable. The discretisation of the advective part of the local transport term in the equations thus requires an upwinding approach. One may chose between a first-order full upwind and a second order exponential fitting scheme, known elsewhere as the Allen-Southwell-Il'in or the Scharfetter-Gummel scheme (Hundsdorfer and Verwer, 2003). It is closely related to the scheme of Fiadeiro and Veronis (1977): on regularly spaced grids both schemes lead to identical discrete forms of the equations. The exponential fitting scheme is, however, better suitable for the flux-conservative finite volume approach on irregularly spaced grids adopted in MEDUSA as it allows for exact mass conservation. Unlike steady-state models, where the solids' advection rate is always oriented downwards relative

to the sediment-water interface, MEDUSA has to be able to cope with solids' advection rates that may have any orientation and that may even change their orientation with time. Both upwinding schemes automatically handle this complication.

At each node (or cell) $j$, the unknowns are the solute concentrations, $(C_i^{\mathrm{f}})_j^n$, the solid concentrations $(C_i^{\mathrm{s}})_j^n$ and the solids' advection rate at the bottom of the cell, $w_{j+\frac{1}{2}}^n$. The advection rate at the top of the cell $j$ is equal to that at the bottom of the cell above $(j-1)$; the advection rate at the SWI is derived directly from the solids' deposition rate, i.e., from the top boundary conditions. The $w_{j+\frac{1}{2}}^n$ at each node are derived from the discrete form of Eq. (3), i. e.,

$$\varphi_{j+\frac{1}{2}}^{\mathrm{s}} w_{j+\frac{1}{2}}^n - \beta_{j+\frac{1}{2}} D_{j+\frac{1}{2}}^{\mathrm{bt}} \left. \frac{\partial \varphi^{\mathrm{s}}}{\partial z} \right|_{j+\frac{1}{2}} = \sum_{i \in \mathcal{I}^{\mathrm{s}}} \vartheta_i (\hat{I}_i^{\mathrm{top}})^n + \sum_{k=T}^{j} h_k \sum_{i \in \mathcal{I}^{\mathrm{s}}} \vartheta_i (\hat{R}_i)_j^n, \tag{7}$$

which furthermore depends on the static volume conservation equation (also at each node)

$$\sum_{i \in \mathcal{I}^{\mathrm{s}}} \vartheta_i (C_i^{\mathrm{s}})_j^n = 1. \tag{8}$$

In Eq. (7), $T$ denotes the top node/cell and the indices $j+\frac{1}{2}$ to $\varphi^{\mathrm{s}}$, $\beta$ and $D^{\mathrm{bt}}$ indicate that these factors are approximations to their respective counterparts at $z_{j+\frac{1}{2}}$.

The complete system of equations is thus overdetermined: at each node, there is one more equation than there are unknowns. However, the equations are not independent of each other. At each node, Eq. (7) is a linear combination of the solids' evolution Eqs. (6) at that node and all the Eq. (7) instances in all the cells on top of cell $j$, furthermore taking Eq. (8) in each of these cells into account. One of the equations is thus redundant at each node. We keep Eq. (7) at each node as it is most convenient to calculate the $w_{j+\frac{1}{2}}^n$ and we furthermore choose to enforce the static volume conservation. Accordingly, one of the solids' evolution equations may be removed at each node to resolve the overdetermination: we choose to drop that for the mandatory inert solid.

## Solution strategy

The discretisation of the DAE system outlined above leads to a coupled system of equations that is generally non-linear due to the expressions for the reaction rate terms $(\hat{R}_i)_j^n$ and needs to be solved iteratively. A full Newton-Raphson approach is unfortunately impractical: due to the Eq. (7) instances in each cell the Jacobian of the complete system is a lower block-Hessenberg matrix which makes the linear system to solve at each iteration computationally expensive. The complete equation system is therefore partitioned in two subsets: the first one with all the Eqs. (7) and the second one with all the remaining equations. Each iteration then proceeds in two stages. First, a fixed-point rule is used to update the advection rate profile (unknowns $w_{j+\frac{1}{2}}^n$) with the first equation subset. The required $D^{\mathrm{bt}}$, $\beta$ and $\alpha$ coefficients and the reaction rate terms are evaluated by using the most recent available concentration profiles (or the initial state). In a second stage concentration profiles (unknowns $(C_i^{\mathrm{f}})_j^n$ and $(C_i^{\mathrm{s}})_j^n$) are then updated by applying a damped Newton scheme (Engeln-Müllges and Uhlig, 1996) for the second subset of equations, using the advection rates calculated at the first stage and considered as constants for this second stage. The algorithm uses the analytical Jacobian which is now block-tridiagonal which allows us to solve the resulting linear system by a block version of the Thomas algorithm. The next iteration then starts again at the first stage, updating the advection rate

profile with the fixed point rule by using the previously updated concentration profiles and then the damped Newton-Raphson correction.

Iterations are stopped on the basis of a two-level criterion, completed by a maximum number of iterations not to exceed (120 by default). It is first required that the Euclidian norm of the scaled residuals of the second subset of equations is lower than $\sqrt{n_C} \times 10^{-6}$, where $n_C$ is the number of equations in the second subset, i. e., the total number of concentration unknowns. Once this first level is reached, we proceed to a root refinement: as long as the maximum number is not exceeded, iterations are continued until the maximum norm of the difference between consecutive scaled concentration iterates falls below $10^{-9}$.

In general, the second level requires only a few extra iterations and may further reduce the equation residuals by several orders of magnitude. Iterations are deemed to have converged once the first-level condition is fulfilled before the maximum number of iterations is exceeded; furthermore reaching the second level is considered a non mandatory extra. The equation system is not explicitly scaled as the linear system solver performs automatic internal scaling (Engeln-Müllges and Uhlig, 1996). The characteristic scales of the components' concentrations, if provided, are nevertheless taken into account in the

convergence criterion and to calculate the equation scales, which are respectively based upon the diffusion time scale of the component whose evolution they describe. For details about the scaling, please refer to the "MEDUSA *Technical Reference*" in the Supplement.

For the initialisation of the iterative scheme, a sequence of approaches has been implemented: (1) the state of the previous time step is used; (2) selected solute profiles are initially set homogeneously equal to the boundary values; (3) a continuation

method where the partial specific volumes of all non-inert solids are gradually increased from zero to their actual values; (4) a continuation method where the top solid fluxes are gradually increased from zero to their actual given values; (5) a continuation method where reaction rates are gradually increased from zero to their standard values; (6) a continuation method only used for steady-state calculations where gradually longer time steps are used and (7) a continuation method only used for columns subject to strong chemical erosion, where the amount of eroded material to return to REACLAY is gradually increased to the

calculated value. These are adopted in turn until one of them leads to a sequence of iterations that fulfils the convergence criterion.

The numerical solution procedure follows an "all-at-once' strategy. Due to the general purpose approach of the model, all chemical reactions are treated equally. It would require artificial intelligence based algorithms to make out efficient processing sequences for arbitrary reaction networks. With fixed compositions and reaction networks, expert knowledge allows the design

of such sequential processing chains, as implemented, e. g., in MUDS (Archer et al., 2002). It is, however, possible to use the code generation facilities of MEDUSA and then to modify the equation solver so that it uses a solution scheme similar to the initialisation strategy (5), but where the reaction rate parameters are not changed homogeneously and continuously, but selectively. Such a modified equation solver would of course be only applicable for that given model configuration.

### 2.2.2    TRANLAY and CORELAY

Once the calculations for one time step in the REACLAY realm have been completed, it is checked whether the sediment column is accumulating ($w_B^+ > 0$) or eroding ($w_B^+ < 0$). If it is accumulating, the mass flux that leaves REACLAY at its bottom

is added to TRANLAY; if it is eroding, then it is furthermore checked whether TRANLAY holds enough material to provide for the calculated influx into REACLAY across its bottom. If it does not, the solid contents of the most recently created layer in CORELAY are returned to TRANLAY and the complete time step is recalculated from the beginning. This is then repeated until

TRANLAY could provide enough material over the whole time step.

At this stage, the concentration and solids' advection rate profiles in REACLAY and the DBL can be accepted for the end of the time step. Finally, the thickness of TRANLAY content is checked: if it is more than 10% thicker than one CORELAY layer (1 cm by default), material for as many CORELAY layers as possible is subtracted and added on top of the current CORELAY stack.

## 2.3    Code Organisation

The MEDUSA common framework includes the subroutines to assemble the equation system and its Jacobian, and to solve the equation system, modules to make available fundamental data (physical constants, unit conversion parameters, etc.), modules to hold the forcing data and the intermediate results. It also provides the core management system for multiple sediment columns, which can be processed in a sequential or a parallel fashion. For parallel processing, Message Passing Interface

(MPI) calls are included and can be activated by a pre-processor switch. Three different MPI interfaces are provided: 1D, 2D and 2D×2D, respectively for a sequential linear distribution of the sediment columns, a two-dimensional ordering (typically longitude-latitude) and a hierarchically ordered two-dimensional array of two-dimensional arrays of sediment columns.

In multi-column setups the chemical composition (solids and porewater solutes) must be the same in all the columns. It is nevertheless possible for each organic matter type (solid or solute) to have different C:N:P:O:H ratios in each column. These

individual ratios must, however, remain constant with time.

The framework system must be completed with the specific parts required for a particular application to build a working instance of MEDUSA. This includes first of all the requested composition in terms of solids and solutes, the reaction network of the diagenetic processes to consider and the chemical equilibria between components of solute systems. MEDUSA must also be aware of the material characteristics such as densities, molar compositions, and some thermodynamic properties for solid

components, or diffusion coefficients for solutes. Rate laws for the different processes under consideration need to be specified.

The information that is required for producing the Fortran code is collected in a series of XML files that define the composition of the sediment and describe the components, processes and equilibria to be considered. These are processed by the MEDUSA COnfigurator and COde GENerator, MEDUSACOCOGEN, to generate as diverse things as modules providing index parameters to address single components by meaningful names, subroutines to calculate molar masses of the components, subroutines

to evaluate reaction rates of all the components and the corresponding derivatives with respect to the relevant component concentrations, to modules to assure the I/O to NETCDF files. The complete diagenesis model is bundled into an object library (`libmedusa.a`) to be linked with the application (host model).

## 2.4 MEDUSACOCOGEN: the MEDUSA COnfigurator and COde GENerator

The "*Reference Guide to the Configuration and Code Generation Tool* MEDUSACOCOGEN" in the Supplement provides an exhaustive description of the procedure to follow to build a working MEDUSA application. The formats of the required files are described in full detail with commented examples. The library of rate law functions and equilibrium relationships is also presented in detail. Further information can be found in the example applications provided in the code. Here, we present only a general overview about the functionality of the code generator.

### 2.4.1 Main Building List

The main building list provides the names of the description files of the solids, solutes and solute systems to consider in a particular model configuration, as well as the process and the equilibrium description file names.

### 2.4.2 Sediment components: solids, solutes and solute systems

Components (solutes or solids) can be of several types and be part of different classes. We distinguish three types of components: *ignored* (default), *normal* or *parameterized* (for solutes only – see below). Evolution equations are only generated for normal components.

*Solids* actually encompass all the characteristics of solid phase components: their concentrations, their age or production time, their isotopic signatures, ... A solid's description file provides information about its physical and chemical properties, such as intrinsic density, alkalinity content per mole (both mandatory), chemical composition, molar mass (optional). A solid can be part of one of four classes: *basic solid* (default), *(particulate) organic matter*, *solid colour* or *solid production time*. The basic solid class includes all physical solids but organic matter, be they reactive or not. For numerical stability reasons, it is mandatory to include at least one inert solid in the model sediment components, to be flagged as mud. There is a dedicated class for organic matter, offering special functionality. Chemical composition is mandatory for this class and can be set in terms of their C:N:P ratios, from which the actual composition is then derived by $CH_2O$, $NH_3$ and $H_3PO_4$ building blocks, or completely in terms of the C:N:P:O:H composition. The solid's colour class can be used for immaterial (volume-less) properties of solids, such as classical colour tracers or isotopic properties. Each component of this class is linked to another solid from which it inherits physical and chemical properties. For components in the solid production time class, the code generator produces adequate equations for age (concentration) tracers. These equations follow the Constituent-oriented Age and Residence time Theory, CART, developed by Delhez et al. (1999) and Deleersnijder et al. (2001). The CART approach provides a means to avoid numerical differentiation in the calculated evolution of the age tracer and the age-carrying component (and also between age tracers carried by different components) as the discrete representation of the underlying evolution equations of the age tracer and the age-carrying component have exactly the same structure and thus suffer from the same numerical dispersion. The original theory has been reformulated here in terms of *production time* instead of *age*. Production time is easier to handle than age in the evolution equations used in MEDUSA. That time remains constant in the absence of chemical reactions and mixing whereas age will continue to evolve with time and thus require a sustained virtual advection

or reaction, even if the material carrying the age tracer gets transferred to the CORELAY realm (where mixing and reactions are ignored). The amended theory is detailed in the technical report "*Early Diagenesis in Sediments. A one-dimensional model formulation*" in the Supplement.

As mentioned above, there is a special type of solutes: *parametrized*. For parametrized solutes, no evolution equations are generated. Their description files must therefore include a code snippet to calculate their abundance or to derive their value

from specific boundary conditions. Typical examples of parametrized constituents are the calcium concentration, which can be derived from salinity, or the saturation concentration of $CO_3^{2-}$ with respect to calcite, which can be calculated from the degree of saturation at the boundary or from the solubility product. The description files of normal solutes must include a code snippet to calculate its diffusion coefficient in free seawater. Solutes' descriptions must also include their specific alkalinity content per mole. We only distinguish between two classes of solutes: *basic solute* (default) and *(dissolved) organic matter*.

Just like particulate organic matter, dissolved organic matter must be characterised by its chemical composition in terms of its C:N:P(:O:H) ratios. For basic solutes, the chemical composition is optional.

Finally, a *solute system* is a set of solutes that MEDUSA considers as a total sum in the equilibrium calculations. Typical examples of solute systems are DIC (dissolved inorganic carbon, composed of $CO_2$, $HCO_3^-$ and $CO_3^{2-}$) or the borate system $(B(OH)_3$ and $B(OH)_4^-)$. Solutes that are part of a solute system are considered to be in local chemical equilibrium with each

485 other. Solute system description files simply list the description files of the solute components that make them up. The code generator internally compiles *Total Alkalinity* as a solute system, based upon the specific alkalinity contents declared in the solute description files included in the model configuration.

### 2.4.3 Processes and equilibria

Description files for *processes* include a representation of the underlying chemical reaction that translates its effect on the

490 various model constituents and specify the rate law to apply. MEDUSACOCOGEN currently provides twenty-one different rate law formulations (actually thirty as several of them have a few variants). Similarly, *equilibrium* description files must include a representation of the chemical equilibrium and specify the law of mass action to use for it. Expressions for laws of mass action (equilibrium relationships) require less variety than process rate laws: the four provided library routines cover the most common cases.

Rate laws and laws of mass action are provided in specially formatted Fortran 95 modules (so-called MODLIB files). The source code of a MODLIB file is preceded by a header (protected by a conditional inclusion pre-compiler directive) that provides meta-data helping to identify and classify the different parameters required. The module itself defines (1) a derived type structure that encapsulates the parameter values and relevant component index references and (2) a subroutine to evaluate the rate-law expression, resp. the equilibrium relationship, for given concentration and parameter values, and the derivatives with

respect to the concentrations of the model components. MODLIB files for laws of mass action must further include a subroutine to set the equilibrium constant (currently derived from the boundary conditions) and another one to evaluate the scale to applied to the equilibrium relationship. The current collection can be easily extended by adding MODLIB files to the library (for details about how to do this, please refer to the MEDUSACOCOGEN reference guide in the Supplement).

Chemical equilibria are always taken into account in the DBL and the REACLAY realms. Processes are usually considered only in the REACLAY realm. However, processes involving only solutes can also be considered in the DBL. In addition, processes that represent radioactive decay of solid trace elements (i.e., of elements whose volume is considered negligible and that do therefore not impinge on the advection rate profile) should be declared to apply also in the TRANLAY and CORELAY realms (where reactions are normally stalled). This way, adequate corrections can be applied in case a sediment column becomes subject to chemical erosion and previously buried material gets remixed into the REACLAY realm. Without such corrections, the material returned to TRANLAY or REACLAY would appear too young.

## 2.5 Code building and customisation options: taming the flexibility

MEDUSA offers a great if not overwhelming deal of flexibility when it comes to setting up, building and running an early diagenesis application. That flexibility begins with the chemical composition (which can be freely chosen, except for a mandatory inert component), the reaction network and the chemical equilibria to consider, the handling of the components (normal or volumeless solids) and the physical processes at work (bioturbation and bioirrigation profiles). It goes on with the configuration of the model domain (its extent, its porosity profile and the resulting tortuosity, with or without a DBL, . . . ) and the numerical (grid layout, upwinding scheme, . . . ) and computational details (debugging output, choice of the coupling API, serial processing or MPI based parallel processing, . . . ).

In the following, I shortly present and discuss a few of the most important of these options. Please refer to the MEDUSACOCOGEN reference guide and the technical report "*Early Diagenesis in Sediments. A one-dimensional model formulation*" in the Supplement for more detailed information about these and further options.

### 2.5.1 Chemical composition and age tracking

The sediment composition, the reaction network and the chemical equilibria to include in the model are obviously the first optional information to decide upon. For applications focusing on a given site or station, these are dictated by available data and observed processes; for MEDUSA applications designed to be coupled to a biogeochemical cycle model, they are defined by the concentration and flux boundary conditions that the host model can provide. When a MEDUSA application module is coupled to a biogeochemical model, it is possible to attach production time to one solid (or even several of them) in order to derive consistent "age models" for the synthetic sediment cores produced, thus providing means for meaningful comparisons to actual sediment core data. It should be noticed though that attaching a time tracer to a solid requires that all the processes relating to that solid have to be duplicated. Since the execution time roughly scales as the square of the number of components, including one or more time tracers may significantly increase the computing time. If such a sediment module is only meant to provide a vertically resolved ocean-sediment exchange scheme, there is no need to include a time tracer.

### 2.5.2 Volumeless solids

The *volumeless solids* option was only introduced to allow the creation of applications that would, as far as possible, be
compatible with other models that do not take the effect of chemical reactions on the advection rate profile into account (e. g.,
Boudreau, 1996; Soetaert et al., 1996; Jourabchi et al., 2008), but that rather link the advection rate profile directly to the
porosity profile via $w(z)\,\varphi^{\mathrm{s}}(z) = w_{\mathrm{SWI}}\,\varphi^{\mathrm{s}}_{\mathrm{SWI}}\;(= w_{\infty}\,\varphi^{\mathrm{s}}_{\infty})$. With this formulation, $w_{\mathrm{SWI}}/w_{\infty}$ is typically of the order of 2–3
whereas it can be easily exceed 10 when the effects of chemical reactions are taken into account. In the test case applications
presented in Sect. 3, the volumeless solids option is only used for the JEASIM application, which replicates a BRNS-global
configuration (Jourabchi et al., 2008) that used such prescribed solids' advection rate profiles. As illustrated in that same
application, this option may lead to physically unrealistic transport. The *volumeless solids* option should therefore only be used
if absolutely required and anyway with great care.

### 2.5.3 Diffusive Boundary Layer

An important option to decide upon is whether a DBL should be included or not. The existence of a DBL at the sea-floor
is merely due to the presence of a more or less sharply defined SWI that delimits the turbulent seawater medium. As such, it
would actually seem a priori indispensable to include a DBL in any model configuration. Interestingly, early carbonate sediment
models, such as that of Schink and Guinasso Jr. (1977), which were essentially designed to allow for an analytical solution,
generally included a DBL. Similar subsequent models (e. g., Keir, 1982; Keir and Berger, 1983) did not include them any
more, possibly as a result of the adoption of non-linear calcite dissolution kinetics, which complicates the integration of a DBL
(Munhoven, 1997). However, even in the later developed complex early diagenesis models, DBLs are not widespread, although
the numerical solution schemes that they use can accommodate that complexity: CANDI (Boudreau, 1996) and OMEXDIA
(Soetaert et al., 1996) are notable exceptions.

The equation system in the continuum part (i. e., REACLAY and DBL) of a MEDUSA column becomes "cleaner" at the SWI
once a DBL is included: without a DBL the SWI lies on a grid node, which is ideal for a prescribed concentration boundary
condition (used for solutes) but less so for a flux boundary condition (used for solids); with a DBL, the SWI lies on a grid
vertex, which is perfect for solids and solutes alike in this case, as both then have to fulfil flux continuity conditions there.
However, integrating a DBL into the model geometry also requires information about its thickness. A DBL acts as a transport
barrier for the exchange of solutes between the sediment and the overlying seawater: the thicker it is, the stronger the resistance
it exerts. For site-specific applications, that information may be derived from solute profiles if they are sufficiently precise.
For global applications the situation is more complicated. DBL thicknesses at the sea-floor range between 100 and 10,000 μm,
with a most probable value close to 1000 μm (Sulpis et al., 2018). A typical thickness of 1000 μm is probably adequate in first
approximation. the effects of a 100 μm and a 10,000 μm thick DBL might, however, be sensibly different.

All in all, it seems recommendable to include a DBL in model setups, especially when information about its thickness is
available. For backwards compatibility with the interfaces to several biogeochemical cycle models that MEDUSA has been

coupled to and that link to MEDUSA's SUBVERSION code repository for this purpose, the default in the code will nevertheless remain "no DBL" for the time being.

### 2.5.4 Other standard options and settings

Most of the options discussed above either directly impinge on the code generated (composition, reaction network, equilibria, volumeless solids, choice of coupling API, selection of MPI processing, . . . ) or have to be selected and defined before compilation (number of nodes for the DBL and REACLAY grids, . . . ). Others can be selected and configured at run-time via specific configuration files, such as the grid point distribution function (six different formulations provided), the porosity profile (two different ones provided), the tortuosity parametrizations (three different ones provided), the biodiffusion constant's profile (seven options), the bioirrigation constant's profile (two options), the upwinding scheme (two options), . . . For several of these, special APIs are furthermore provided to manage custom formulations.

## 3 Test Case Applications

Three applications have been selected to illustrate the functionality of MEDUSA: (1) a replication of the ALL simulation experiment from Munhoven (2007), supplemented with an extended configuration that attaches an age tracer to calcite; (2) a coupling simulator where the boundary conditions that would normally be provided by a biogeochemical model that MEDUSA would be coupled to are read from a file; (3) a MEDUSA configuration with complex composition and reaction network as considered in state-of-the-art early diagenesis models used for the analysis of site observational data. All of the test case applications use $\alpha \equiv 0$ (no bioirrigation).

### 3.1 MEDMBM-PT – Coupled simulation experiment with chemical erosion and resolved sedimentary records

MEDUSA produces truly resolved and, via its CART-based age/production time control system, consistently dated synthetic sedimentary records. In order to illustrate the potential of this combination of features, the ALL experiment from Munhoven (2007) is repeated here with a MEDUSA configuration equivalent to the one used in that study, extended to attach age control information to calcite particles.

#### 3.1.1 Application description

Munhoven (2007) used MBM (Multi-Box-Model) coupled to MEDUSA-v1 to explore the implications of the rain ratio changes proposed to explain the glacial-interglacial atmospheric $CO_2$ variations (see, e.g., Archer and Maier-Reimer, 1994) for the preservation-dissolution pattern of carbonate over these time scales. MBM is an eleven-box model of the carbon cycle in the ocean and the atmosphere. The ocean is subdivided into ten reservoirs as a function of depth (surface, intermediate and deep layers), latitude (low-latitude, northern and southern high latitudes) and ocean basins (Atlantic, Antarctic and Indo-Pacific). The ocean reservoirs each have a depth distribution derived from the hypsometric curve. MBM calculates the evolutions of DIC, TA, phosphate (the limiting nutrient) and oxygen in the oceanic reservoirs and $pCO_2$ in the atmosphere. In addition,

$\delta^{13}C$ and $\Delta^{14}C$ are considered for all carbon bearing tracers and fluxes. Organic carbon, calcite and aragonite are produced by biological activity in the surface reservoirs and settle down to the intermediate and deep reservoirs. Part of the organic matter gets remineralised in the water column, the rest transferred to the sediment; all of the carbonate rains down to the sea-floor and enters the sediment. The coupled model includes 304 MEDUSA sediment columns with a 10 cm thick bioturbated surface mixed-layer, distributed as a function of depth (one column for each 100 m depth interval of sea-floor) over the five

ocean basins delimited by the five surface reservoirs (North Atlantic, Equatorial Atlantic, Antarctic, Equatorial Indo-Pacific and North Pacific). A grid with 21 nodes was used for the sediment columns. The sediment composition is the same as in Munhoven (2007): the solid phase is composed of clay, calcite, aragonite and organic matter; pore water solutes are $CO_2$, $HCO_3^-$, $CO_3^{2-}$ and $O_2$ as solutes, the carbonate system being in thermodynamic equilibrium. Here, we further include an age tracer, carried by calcite. Processes taken into account are calcite and aragonite dissolution as well as oxic organic matter

remineralisation. We make the simplifying assumption that the particles' age does not influence its solubility, which allows for a straightforward generation of the required evolution equation terms by MEDUSACOCOGEN. The exact formulations of the rate laws and the adopted parameter values are given in Table 3. MBM furthermore considers carbonate accumulation on the continental shelf.

The main driving forces in the ALL experiment are (1) the changing shelf accumulation rates, which depend on the extent

of low-latitude flooded shelves, which depends in turn on the sea-level whose history is prescribed, (2) the changing export rain ratio (i.e., the carbonate-C to organic-C in the biogenic export fluxes), whose evolution is prescribed as well and which is 40% lower at peak glacial than at peak interglacial times and (3) the riverine $HCO_3^-$ inputs and atmospheric $CO_2$ consumption rates, which are derived from Jones et al. (2002). The deep-sea sedimentary accumulation patterns adjust onto the DIC and TA variations induced by these three factors. Please refer to Munhoven (2007) for additional details and references.

### 3.1.2 Results and discussion

Simulation experiments were run over 240,000 yr with cyclically repeated forcings (temperature, sea-level, rain-ratio changes, weathering, etc.) with a 120,000 yr period. Figure 2 shows two aspects of the surface sedimentary $CaCO_3$ content. The top panel depicts the actual time-dependent evolution of that content in the sedimentary mixed-layer in the Equatorial Indo-Pacific part of the model ocean (indistinguishable from that shown in Munhoven (2007)). The bottom panel shows the resulting

sedimentary record (synthetic cores), where each dot represents a 1 cm thick sample and the crosses depict the composition of the surface mixed layer (the REACLAY realm) as a function of depth, most of which overprint each other as both age and material distributions are rather homogeneous there, especially at depths shallower than 4700 m.

It should be noted that the black dots, which correspond to samples almost devoid of the time-carrying calcite component, provide only incomplete or unreliable information and several of these may possibly overprint each other. The white and grey

lines in the top panel respectively represent the evolutions of the carbonate compensation depth (CCD, defined here as the depth where the carbonate dissolution rate is equal to the carbonate deposition rate) and of the calcite saturation horizon (CSH, i.e., the depth at which saturation with respect to calcite is reached). The two distributions are broadly similar but present noticeable differences. There is first of all a systematic time lag of about 3 to 8 kyr, best discernible in the regions where

%CaCO$_3$ exceeds 70%. This time lag is due to the fact that the age tracer tracks the average age of the surface sediment at burial, which is different from zero at all times. Another important difference is the alteration of the actual evolution (Fig. 2a) in the sedimentary record (Fig. 2b). The most striking differences are visible during times of shoaling CCD at depths where %CaCO$_3$ is lower than 70%. As shown by the long-dashed line on Fig. 2b, which traces the evolution of the limit between the maroon and the black zones from Fig. 2a (the 0% calcite line), shifted by 5 kyr towards the greater ages to account for the average calcite burial age, up to 20 kyr of calcite history have been deleted from the record between 4700 and 6700 m d.b.s.l. due to chemical erosion.

MEDUSA always includes at least internally a TRANLAY-CORELAY stack of sediment layers underneath the REACLAY realm in order to handle possible chemical erosion events. These layers can a priori be only approximately dated from the recorded time of burial for each layer, i.e., the time when a given layer is separated from TRANLAY and transferred to CORELAY. With the CART based production time information attached to a solid constituent, this shortcoming can be overcome.

## 3.2 COUPSIM – Coupling simulator

MEDUSA has already been coupled to several ocean biogeochemistry and Earth System Models. First results have been published for the coupling to the Community Earth System Model, CESM (Kurahashi-Nakamura et al., 2020); other coupling projects are well advanced (Moreira Martinez et al., 2016; Völker et al., 2020).

To illustrate the procedure of coupling MEDUSA to a biogeochemical model and to assess the computational cost of a typical sediment module for a real three-dimensional ocean biogeochemistry model, a coupling simulator, COUPSIM, was developed where the boundary conditions that would normally be provided by the host biogeochemical model are read from a file.

### 3.2.1 Application description

We use results obtained with the coupled Biogeochemistry/Ecosystem/Circulation model, BEC (Moore et al., 2004),[3] that were made publicly available (Moore et al., 2005) in the framework of the Synthesis and Modelling Project of the U.S.-Joint Global Ocean Flux Study (U.S.-JGOFS) research program. That version of BEC consists of a marine ecosystem model coupled to a preliminary version of the Community Climate System Model, CCSM 2.0, Parallel Ocean Program, POP. The coupling simulator was designed to run with annual or multi-annual time steps. The required forcing data were therefore extracted from the provided monthly-mean datasets and aggregated into a yearly average climatology.

A MEDUSA configuration with four solid (clay, calcite, organic matter and opal) and six solute components ($CO_2$, $HCO_3^-$, $CO_3^{2-}$, $O_2$, $NO_3^-$ and $H_4SiO_4$) was chosen. The processes considered are calcite dissolution, oxic respiration of organic matter, organic matter degradation by nitrate reduction and full denitrification and opal dissolution. The exact rate law expressions used are given in Table 4. The model sediment columns are supposed to extend over the typical bioturbated mixed-layer depth of about 10 cm throughout, covered by a grid with twenty-one nodes. The vertical resolution of the sediment column is thus of the same order as that of the water-column in BEC which has twenty-five vertical layers. MEDUSA sediment columns were attached to sea-floor grid elements at depths greater than 1000 m below sea-level, which amounted to 7332 globally. COUPSIM works

---

[3]Later references (e.g., Moore et al., 2013) resolve BEC as "Biogeochemical Elemental Cycling."

perfectly well at shallower depths,[4] but the BEC boundary conditions lead to completely unrealistic sediment compositions there, such as organic carbon contents of 40% and more over widespread areas. The BEC set-up used by Moore et al. (2004) calls upon a reflective boundary condition at the sea-floor: all of the biogenic material that reaches the bottom-most ocean cells gets entirely remineralised there. Shortcomings of that approach have already been mentioned in the introduction. Here, we have to add another important one: remineralisation or dissolution rates which normally change gradually with depth in the water column present a sudden increase from the second deepest to the deepest layer, often by an order of magnitude, in some instances even by two or more. The dissolution flux in the bottom cells thus has two contributions: one part stems from the dissolution in the water column covered by the bottom cell and one part is related to the reflective boundary condition. To separate these two parts, we therefore extrapolate the water column dissolution flux profile to the bottom cells assuming an exponential decrease with depth, using the values in the two ocean cells overlying each ocean bottom cell. The total bottom dissolution flux is then corrected for this water column dissolution part and the rest is supposed to settle at the sea-floor where it enters the surface sediment.

### 3.2.2   Results and discussion

Reaction rate constants for the different processes have been adjusted in order to derive surface average sediment compositions that come as close as possible to observed distributions. For this adjustment process, the values of the parameters in the dissolution and remineralisation rate laws were varied and the model run to steady-state for each parameter set.

The resulting global distributions of the solid fractions of calcite, total organic carbon and opal (resp. denoted %Calcite, %TOC and %Opal hereafter) were then evaluated against observational data (Seiter et al., 2004) on the basis of their respective standard deviations and correlation coefficients. The results of these experiments are summarised in the Taylor diagram (Taylor, 2001) on Fig. 3. For that diagram each one of the three distributions was normalized with respect to the standard deviation of its respective observational counterpart in order to be able to report all the results on a common scale. The peculiar distributions of the different characteristic points on that diagram indicate that there is an structural incompatibility between the data and the possible model results. Calcite points cluster around a correlation coefficient of 0.38 and a standard deviation of 1.3, despite a large range of values ($5$–$1000\,\%\,\mathrm{day}^{-1}$) used for the dissolution rate constant ($k_\mathrm{c}$ in Table 4) in the experiments. The opal and organic carbon points align on two beams with correlation coefficient values between 0.23 and 0.27, and between 0.35 and 0.45, respectively, each one with a large range of standard-deviations. For the organic matter dissolution rate laws values between $0.005$ and $0.032\,\mathrm{yr}^{-1}$ were adopted independently for the two rate constants ($k_\mathrm{ox}$ and $k_\mathrm{nr}$ in Table 4); for opal dissolution, rate constant values between $0.04$ and $0.07\,\mathrm{yr}^{-1}$, together with asymptotic concentrations ranging between 500 and $700\,\mathrm{\mu mol\,L}^{-1}$ were used (resp. $k_\mathrm{o}$ and $C_\mathrm{os}$ in Table 4).

Among all the experiments carried out, the one that offered the best compromise in terms of standard deviations coming as close as possible to the observations (i.e., to 1 after the normalization) and maximizing the correlation was selected as the best-fit experiment. The corresponding results are represented by the full symbols on Fig. 3. That experiment is characterised by a calcite dissolution rate constant of $36.525\,\mathrm{yr}^{-1}$ (i.e., $10\,\%\,\mathrm{day}^{-1}$, only 1/10th of the $100\,\%\,\mathrm{day}^{-1}$ of Archer (1991))

---

[4]Please see the "Reply on RC1" in the GMDD version of this paper (doi:10.5194/gmd-2020-309-AC1) for instructions about how to verify this.

for calcite dissolution. For organic matter remineralisation the best-fit rate constants are $0.015\,\mathrm{yr}^{-1}$ and $0.005\,\mathrm{yr}^{-1}$ for oxic respiration and the oxidation by nitrate reduction, respectively. These values compare well with the results of Palastanga et al. (2011), who also adopt a 1G approach as we have done here, but use different rate constants at depths shallower than $2000\,\mathrm{m}$ ($k_{\mathrm{ox}} = 0.01\,\mathrm{yr}^{-1}$ and $k_{\mathrm{anox}} = 0.008\,\mathrm{yr}^{-1}$) and greater than $2000\,\mathrm{m}$ ($k_{\mathrm{ox}} = 0.005\,\mathrm{yr}^{-1}$ and $k_{\mathrm{anox}} = 0.002\,\mathrm{yr}^{-1}$). For opal, a comparatively high value of $0.05\,\mathrm{yr}^{-1}\,(\mathrm{mol\,m}^{-3})^{-1}$ had to be adopted in order to avoid widespread opal-dominated sea-floor sediments. For comparison, the rate constant of $30\,\mathrm{yr}^{-1}$ of Boudreau (1990) translates to $0.0034\,\mathrm{yr}^{-1}\,(\mathrm{mol\,m}^{-3})^{-1}$ for the rate-law formulation adopted here. The asymptotic ("saturation") concentration for opal dissolution had to be set to $700\,\mu\mathrm{M}$, which ranges at the cold-water end ($\leq 1\,^{\circ}\mathrm{C}$) of the experimentally derived values of Dixit et al. (2001).

The resulting average surface model sediment composition at steady state is compared to the target data of Seiter et al. (2004) on Fig. 4. Calcite-rich sediments are produced in the Indian and South Pacific; the calcite-rich sediments along the Atlantic mid-ocean ridge are only reproduced in the South Atlantic and at mid-latitudes in the North Atlantic. The sediments that are richest in TOC are located in the equatorial East Pacific. The opal belt in the Southern Ocean stands out, as well as the maximum in the equatorial East Pacific. This latter is, however, too narrow and similarly to the calcite maxima too sharply delimited.

Among the structural incompatibilities, the band of calcite-rich sediments along the entire rim of the Pacific Ocean in the Northern Hemisphere stands out. These are not seen in the data for the simple reason that the CSH is actually shallower than $1000\,\mathrm{m}$ almost everywhere along this band, except in the Sea of Japan (or East Sea) (Yool et al., 2001). In the BEC results, this whole band is supersaturated with respect to calcite. No parameter combination can override this supersaturation and significantly reduce the amount of carbonate preserved. The reflective boundary condition actually contributes to this unrealistic supersaturation, possibly even causes it: as all of the calcite that reaches the deepest ocean cells there unconditionally dissolves the degree of super-saturation of the bottom waters is artificially increased. In the coupled CESM-MEDUSA model experiment of Kurahashi-Nakamura et al. (2020), this carbonate-rich band is not produced. Another important feature of the model %Calcite is the absence of carbonates at the sea-floor in the Atlantic north of the equator and south of $30\,^{\circ}\mathrm{N}$, again in contradiction with the data. This results from very low $\mathrm{CaCO_3}$ deposition rates of the order of $1\text{–}3\,\mathrm{g\,m}^{-2}\,\mathrm{yr}^{-1}$, combined with high lithogenic deposition rates of the order of $15\,\mathrm{g\,m}^{-2}\,\mathrm{yr}^{-1}$ in the BEC results, which caps %Calcite at about 5–15% a priori even in the absence of dissolution. The extended areas of carbonate-rich sea-floor sediments along the North Pacific rim and of strongly diluted sediments in the central Atlantic contribute to the poor correlation coefficient, whatever the model parameter combinations chosen.

In general, the model sediment compositions show far more pronounced contrasts than the observed ones. In all three distributions, the more diffuse features of the global distributions seen in the data are missing. Intermediate values are widespread in the observed distributions, especially for organic carbon where there are only a few isolated spots at the upper end of the depicted range. The model produces in most regions surface sediments that are either poor or extremely rich in calcite. There are only few areas with calcite fractions between 20 and 80 %. The same holds for organic carbon and opal, albeit to a lesser degree for the latter which presents more extended areas with intermediate abundances.

The sharp contrasts for %Calcite can partly be explained by to the vertical grid resolution of the BEC version used by Moore et al. (2004). The CSH in the South Pacific is situated at a depth of about 2.5–3.3 km and at about 3–3.7 km in the Indian Ocean (Yool et al., 2001). At these depths, the vertical resolution of BEC is about 340–450 m. As the transition zone from carbonate-rich to carbonate-poor sediment depths is about 500 m thick there, it is clear that this transition zone cannot be satisfactorily resolved. There are several options to alleviate this shortcoming. The most obvious one is of course to increase the vertical resolution of the host model. The CESM version used by Kurahashi-Nakamura et al. (2020) has sixty vertical levels. Another option would be to call upon sub-grid scale depth profiles and to attach several (two to four) sediment columns to each model sea-floor grid element. This would of course increase the computational burden of the sediment part of the model, but as can be deduced from the execution times reported and discussed below, these would not represent a major hindrance, given the efficiency of the numerical procedures adopted in MEDUSA.

The %Opal distribution is essentially correlated to the deposition flux rates of opal, which themselves present a highly contrasted distribution. %TOC appears to depend on the organic deposition flux rate and the bottom water oxygen distribution. The preservation tongue in the equatorial East Pacific results from a well delimited deposition rate maximum, in combination with mid to low oxygen concentrations ($20$–$160\,\mu$mol L$^{-1}$). The high organic deposition rates in the Southern Ocean and along the west coasts of Africa do, on the other hand, not lead to high %TOC as these regions are much better oxygenated ($160$–$250\,\mu$mol L$^{-1}$).

### 3.2.3 Execution times

The execution times for the best-fit experiment for different computing environments (serial, parallel MPI with the MPI-1D and MPI-2D interfaces) are reported in Table 5. MPI-2DT2D execution times are not reported as they are always within a few percent of those for MPI-2D. The results discussed above were for steady state, i.e., one infinitely long time step. To assess the computational overburden of typical experimental runs, we further added execution times for a 1000 yr simulation experiment, with sediment model steps ranging from 1 to 1000 yr. To allow for an easier comparison, the execution times are reported relative to the serial experiment with 100 time steps of 10 yr which took 9:14.73 min on the computing platform used for the experiments. The usage of one-year time steps makes this serial experiment last 7.7 times longer (1:11:09 hours). Parallel execution allows a considerable reduction of these execution times: with two processors, the execution time of the reference experiment is reduced by 38% and by 68% with four processors (using the MPI-2D interface). Relative reductions are nearly the same for the simulations with a one-year time step. The MPI-1D, which offers the best work-load balance in theory as the sediment columns can be optimally distributed among the processes, is about 10–20% less efficient than the MPI-2D interface, which offers a poorer work-load balance in theory, because of a more complicated coordination overhead. All in all, we may expect that the computational overburden of MEDUSA would represent only a small fraction of the total execution time of a complex Earth System Model.

### 3.3 JEASIM – Complex composition and reaction network model

In order to compare the performance of MEDUSA to that of other state-of-the-art early diagenesis models, we revisited the study of Jourabchi et al. (2008) who used the Biogeochemical Reaction Network Simulator, BRNS (Jourabchi et al., 2005) to analyse sea-floor sediment $O_2$ and $pH$ profile data from thirteen different sites. The model description in Jourabchi et al. (2008) is sufficiently detailed and complete to allow a meaningful replication.

#### 3.3.1    Application description

Here we simplify the original model configuration by neglecting the sulfurous constituents (sulfate and sulfide) and methane as well as the sulfur-based processes (sulfate reduction, sulfide oxidation) and methanogenesis. Jourabchi et al. (2008) found that these were significant at a few sites only (their sites 40, 48 and 120). We furthermore do not explicitly consider $H^+$ and $OH^-$ but similarly to Van Cappellen and Wang (1996), we include their effects implicitly into the combined carbonate and borate equilibria. The adopted model configuration thus includes $CO_2$, $HCO_3^-$, $CO_3^{2-}$, $O_2$, $NO_3^-$, $Mn^{2+}$, $Fe^{2+}$, $NH_4^+$, $B(OH)_3$ and

$B(OH)_4^-$ as solutes, two classes of organic matter, clay, calcite, $MnO_2$ and $Fe(OH)_3$ as solids. The primary redox reactions considered are organic matter oxidation by oxic respiration, by nitrate reduction and denitrification, by Mn(IV) reduction and by Fe(III) reduction, each one duplicated for the two classes of organic matter. The secondary redox reactions considered are nitrification, $Mn^{2+}$ and $Fe^{2+}$ re-oxidation by $O_2$ and $Fe^{2+}$ re-oxidation by $MnO_2$. Furthermore, calcite dissolution is included. The carbonate and borate systems are kept in equilibrium. The resulting application is called JEASIM (Jourabchi Et

Al. SIMplified).

     Unlike BRNS-global, MEDUSA is based upon a complete composition approach with regard to solids: it is assumed that the composition of the solid phase is completely known and the solids' advection rate profile is deduced from the porosity and the reaction rate profiles (see Eq. (3)). Jourabchi et al. (2008) do not consider the effect of chemical reactions on the solid advection rate profile but only set the burial velocity, from which the advection rate profile is then derived, assuming that reactions do

not have any influence on the advection rate profile (i.e., the integral term in Eq. (3) is neglected). This behaviour can be simulated in MEDUSA by calling upon the *volumeless solids* option at compile time. With this option, only the main inert solid is considered to have a finite density. All the others are considered to be tracers without significant volume, but only mass (i.e., they have zero partial specific volume and infinite density). Both approaches are mathematically equivalent and both have an important drawback: they allow to transport physically unrealistic amounts of non-inert material (e. g., calcite mass fractions

exceeding 100%), as we will see below. The model sediment columns were assumed to extend down to $82\,\mathrm{cm}$ as in the original study; we used a REACLAY grid with 321 nodes, without a DBL. Porosity and bioturbation profiles were prescribed, using the information provided by Jourabchi et al. (2008). Boundary and forcing conditions (solute concentrations, calcite saturation state at the SWI, solids' burial rate) were also taken from Jourabchi et al. (2008). For our purpose, the solids' burial rate was converted to an equivalent flux of inert material (dubbed clay) across the SWI.

### 3.3.2 Adjustment procedures

The adjustment was carried out in two stages, for each of the thirteen stations. At the first stage, the organic matter deposition rate and the degradation rate constants were adjusted in order to reproduce the measured oxygen concentration profiles; calcite dissolution was ignored. For the second stage, these two parameters were held fixed at the values obtained during the first stage. This time, the calcite deposition rate and the calcite dissolution rate constant were adjusted in order to reproduce the measured $p$H profiles. We only considered calcite dissolution rate orders 1, 2 and 4.5, disregarding the relatively uncommon order 0.5. In contrast to Jourabchi et al. (2008), we did not adopt static initial values for the calcite dissolution rate constant at the second step. We tested random perturbations to these in order to allow a deeper exploration of the results space. In a few instances, this allowed us to significantly improve on the results of Jourabchi et al. (2008). Fits were carried out with MPFIT (Markwardt, 2009)[5] with the GNU Data Language (GDL, v. 0.9.6). The minimisation procedure implemented in MPFIT is based upon the Levenberg-Marquardt algorithm. For the $p$H profile fitting, calculations were based upon $H^+$ concentrations derived from the $p$H data.

### 3.3.3 Results and discussion

O$_2$ data are comparatively straightforward to fit. The model profiles obtained are essentially identical to those of Jourabchi et al. (2008) and we are therefore not discussing them here. The detailed results can be found in the `jeasim_vl.ods` spreadsheet file in the `work/jeasim` directory of the code and data archive provided in the Supplement, on the sheet named `jeasim_adj0`; the resulting O$_2$ profile fits are shown on Figs. S1 and S2 in the "*Additional Results*" in the Supplement. In terms of the relative misfits, the quality of the fits obtained here is even in most instances slightly superior to those of Jourabchi et al. (2008), except for site 2, where ours is 37% worse. Three others (sites 19, 40 and 59) have similar relative misfits (within $\pm 10\%$ of those of Jourabchi et al. (2008); all the rest present between about 20 and 80% lower relative misfits.

$p$H profile adjustments, on the other hand, were far less straightforward. We therefore proceeded in several steps (up to three). First, the fitting procedure was carried out with only the $p$H profile data as a constraint. As mentioned above, the way the application is designed (with the *volumeless solids* option), physically unrealistic mass fractions exceeding 100% cannot be precluded a priori. The fits for six out of the thirteen stations considered (shown on Fig. 5) were affected by this shortcoming. For these stations, the optimal calcite fraction in the total mass entering the sediment column at the SWI ranged between 140 and 5400% (see Table S1 in the "*Additional Results*" in the Supplement). At sites 19 and 20, the resulting calcite mass fraction in the surface sediment solid fraction was nevertheless physically acceptable; at the four other stations it ranged between 330 and 4100%. For sites 49 and 57 the optimal $p$H profiles are in poor agreement with the data. The same was observed by Jourabchi et al. (2008): for site 49, they did not even report any $p$H results and for site 57, they only obtained a profile for the dissolution rate order 0.5, not considered here. We have been able to derive fits for site 57 for each rate order considered. All of these strongly resemble the profile of Jourabchi et al. (2008) for $n = 0.5$ and also fit the data in a similarly poor way.

---

[5]Available from http://purl.com/net/mpfit.

For these six sites a second round of fits was performed where the calcite mass fraction in the total deposition flux at the SWI was limited to 90%. The $p$H profiles from these second fits (except for site 49 which we do not consider any further) together with those of the other sites that passed the first step are shown in Fig. 6. Sites 19 and 20 are again noteworthy as the calcite input limitation does not allow to obtain any meaningful $p$H profile to fit the data for these. It should, however, be noticed that Jourabchi et al. (2008) also obtained only physically unrealistic results for sites 19 and 20: the calcite deposition flux rates reported for these two sites in their Table 8 would actually require minimum advection rates at the SWI that exceed the prescribed ones by factors of 2.7 to 4.7. For sites 40 and 48, the $p$H profiles found here are essentially indistinguishable from those of Jourabchi et al. (2008) (although the profiles found for $n = 1$ and $n = 4.5$ at site 48 appear to be swapped). For sites 39 and 58, we find less pronounced subsurface minima than Jourabchi et al. (2008): at site 39, the fits for orders 1 and 2 are improved here but degraded for order 4.5; at site 58 all fits are better here as the data do not support well-pronounced subsurface minima. The optimized $p$H profiles obtained here yield better fits at site 41 for $n = 1$ and $n = 2$, but a worse one for $n = 4.5$. The profile for $n = 4.5$ exhibits too strong a subsurface minimum together with too high $p$H values at depth. However, attempts to improve the fit of the subsurface minimum deteriorates the quality of the fit at depth, increasing the overall relative misfit. All $p$H profiles derived here at site 50 are similar to those obtained by Jourabchi et al. (2008) or better. In particular, the profile for $n = 4.5$ does not present a systematic bias towards lower $p$H values. At site 59, we obtain significantly better $p$H profile fits. For that site, Jourabchi et al. (2008) had only found $p$H gradients at the SWI that were opposite to the observations with their model, whereas the gradients obtained here are in agreement with the data (for all calcite dissolution rate orders). For site 120, finally, our $p$H profiles do not present the bias towards low $p$H witnessed by the results of Jourabchi et al. (2008). They rather present a bias towards higher $p$H, caused by the three outliers between 2.5 and 3 cm depth: additional experiments (not shown) where these outliers were removed allowed to improve the relative misfit by a factor of four. In addition, we have also been able to produce a $p$H profile for $n = 4.5$.

Upon inspection of the other characteristics of the resulting model sediment, we found that most of the sites presented far too low calcite mass fractions in the surface sediment (see Table S2 in the "*Additional Results*" in the Supplement). Only at sites 39 and 57 does it come close to the observed 90%. This is, however, most likely due to the imposed deposition flux with 90% calcite at these sites. In a third step, we have therefore repeated the experiments for all the sites by adding the surface sediment calcite mass fraction as constraint, letting the mass fraction of calcite in the deposition flux float again. For this purpose, the calcite fractions reported by Jourabchi et al. (2008, Table 1) were assumed to hold for the average over the top 10 cm of the sediment. The resulting $p$H profiles are shown on Fig. 7. The profiles obtained for sites 2, 19 and 20 require, once more, a few comments. They all apparently present more or less acceptably fitted $p$H profiles, only slightly worse than at the first step (Fig. 5), but far better than at the second step (Fig. 6). However, the constraint on the surface sediment mass fraction of calcite is unfortunately not sufficient to yield physically realistic results at these three sites. The calculated optimal calcite mass fractions in the deposition flux for all dissolution rate orders range from 140 to 730%, except for site 2, where $n = 1$ and $n = 2$ provide acceptable results, but not $n = 4.5$ (see Table S3 in the "*Additional Results*" in the Supplement). There is little difference between the fits with the limited calcite deposition flux at sites 39, 48, 50, 57, 58 and 120. At site 59, for which the optimal $p$H profiles obtained at the previous step were superior to those obtained by Jourabchi et al. (2008), the gradients

now change sign for $n = 1$ and $n = 2$ and the quality of the fits degrades; the quality of the profile obtained for $n = 4.5$ is also sightly degraded but remains acceptable. At site 41, finally, the differences are comparatively large, except for $n = 4.5$. The reason for these contrasting changes are simple. The target %Calcite is 87%. During the first fitting exercise, the %Calcite was 77% for $n = 4.5$, but only 0.04% and 0.06% for $n = 2$ and $n = 1$, resp. Accordingly, these latter cases require strong increases in the calcite deposition rate combined with strongly reduced dissolution rates to meet that imposed constraint. All in all, we find that the two models produce results in agreement with each other. Interestingly, the stations for which Jourabchi et al. (2008) diagnosed a significant role for sulfur-based processes yield optimal $pH$ profiles here that are essentially identical than the original ones. This should, however, not be seen as a proof that sulfate reduction and subsequent sulfide oxidation are negligible at the sites under study. It should indeed not be forgotten that the $O_2$ and $pH$ data profiles only cover the uppermost parts of the sediment porewaters, in some instances only the top-most 1–2.5 cm: for $O_2$ these are the profiles at sites 2, 19, 20, 48 and 50, for $pH$ those at sites 19, 20, 48 and 50. They do thus provide only a weak constraint on the deeper parts of modelled sediment depth, where sulfate reduction fuelled processes typically take place.

Although it would certainly be interesting to investigate the reasons for the failure to produce satisfactory fits to the $pH$ data profiles with our models, this would go beyond the scope of this model description paper. Besides the simplifications in the sediments compositions and the reaction network, there remain a few differences between BRNS-global and the JEASIM application of MEDUSA and between the adopted fitting procedures. We believe that the differences between the two models only have minor impact. BRNS-global takes the porewater advection into account, whereas MEDUSA does not by default (as adopted here), since it is small compared to diffusive transport. The fitting procedures may have a more important impact. The optimisation routine used by Jourabchi et al. (2008) calls upon a simplex algorithm; MPFIT uses a Levenberg-Marquardt method and furthermore allows to take range constraints for the solutions into account. Both algorithms are iterative and suffer from inherent shortcomings. It is well-known that the simplex method may converge to non-stationary points. With MPFIT we have experienced situations, where we had to increase the a priori physically sufficient lower boundary of zero to some non-zero positive value. The iterations eventually "bounced" upon this non-zero value and made the procedure converge to some well-pronounced minimum, whereas the theoretically sufficient bound of zero would have made the iterations converge to some random low value, producing $\chi^2$ values far above the potential minimum. Furthermore, it should be noticed that we have carried out the $pH$ fitting on the corresponding $H^+$ concentrations, whereas Jourabchi et al. (2008) used the $pH$ values instead. $H^+$ concentrations offer a far larger dynamical range than $pH$, thus providing a stronger constraint.

We concur with Jourabchi et al. (2008) in concluding that there are probably other constituents, reactions and processes not considered in our models that may significantly impinge on the proton exchange reactions and thus influence the $pH$ profiles in a way that can thus not be reproduced. However, there are additional assumptions that must also be considered to explain the model-data discrepancies. The empirical relationship used to derive the burial rate from the sea-floor depth is possibly unreliable for usage at isolated sites. At some stations, a model configuration with a DBL might be required. This is almost certainly the case at site 120, for which Wenzhöfer et al. (2001) have made out a DBL thickness of $725 \pm 25\,\mu m$.[6]

---

[6]Site 120 is identical to site GeoB4901 in Wenzhöfer et al. (2001).

## 4 Conclusions

In its version 2 MEDUSA, the Model of Early Diagenesis in the Upper Sediment with Adaptable complexity, offers a flexible approach to develop tailored sediment modules suitable for coupling to ocean biogeochemical cycle models, but also for process analysis and teaching. MEDUSA is based upon the standard time-dependent diagenetic equation (e.g., Berner, 1980; Boudreau, 1997) with the vertical as the only spatial dimension. The chemical composition of the solid and solute phases can be chosen to fit the requirements of the scientific question to address. The network of diagenetic reactions (redox processes, mineral dissolution, ...) and chemical equilibria in the porewaters can be made exactly as complex as required. The Application Programming Interfaces (APIs) allow the coupling to the most commonly encountered geographical grid layouts in biogeochemical models (unstructured, two-dimensional, or hierarchically ordered two-dimensional arrays of two-dimensional areas). They have furthermore been designed so that existing model configurations can be easily extended (or simplified). The adopted numerical procedures allow for large range of time steps, making MEDUSA suitable for long simulation experiments with well-resolved representations of the sea-floor (several thousands of sediment columns). Parallel processing for multi-column set-ups is supported via Message Passing Interface (MPI) instructions that can be optionally activated.

Three test case applications have been selected and presented in this model description paper. The first one revisits the study of Munhoven (2007) which used the predecessor version MEDUSA-v1. A model configuration with exactly the same functionality as MEDUSA-v1 was set up and extended by the age monitoring facilities offered by MEDUSA. These latter are based upon the Constituent-oriented Age and Residence time Theory, CART (Deleersnijder et al., 2001; Delhez et al., 1999), which has a well-established record in the analysis of geophysical fluid flow models. CART can be easily transposed to the standard equations describing the early diagenesis of sea-floor sediments. The newly generated model code accurately reproduces the original results. The added age-control information provides insight into the relationship between the actual time-dependent evolution of the sea-floor sedimentary mixed-layer composition and the resulting sedimentary record.

For the second application, a coupling simulator was developed to illustrate the opportunities and to assess the computational requirements of using a fully coupled vertically resolved early diagenesis model as the ocean-sediment exchange scheme in a model of ocean biogeochemical cycles. MEDUSA offers attractive flexibility requiring reasonable computational overburden. MPI interfaces allow for a significant gain in computing time. Even fully coupled simulation experiments with a horizontal resolution of $1° \times 1°$, with of the order of 40,000 sea-floor grid points are conceivable. MEDUSA is thus well suited to upgrade commonly used category 1, 2 and 3 ocean-sediment exchange schemes in biogeochemical models.

The third application considered a complex composition and reaction network. We analyse and compare the performance of MEDUSA to that of the state-of-the-art early diagenesis model BRNS-global (Jourabchi et al., 2005) by revisiting the study of Jourabchi et al. (2008). In that study, porewater $O_2$ and $p$H microelectrode profiles taken in situ were interpreted with BRNS-global. The results obtained with MEDUSA compare favourably to those obtained by Jourabchi et al. (2008).

Like most models MEDUSA has, and will probably always have, a touch of a "never-ending story." In one of the next stages, adsorption processes will be added to MEDUSACOCOGEN. The currently generated model code can be manually amended or patched to take into account adsorption along the lines developed by Berner (1976). A consistent and mechanistically based

approach with as few early-stage simplifying assumptions as possible would nevertheless be preferable and a treatment similar to the solute systems is currently being worked out. It is furthermore planned to improve the computing time requirements by dealing separately with variables that do not have any impact on the solids' advection rate profiles. In general, isotopic signatures, production times, colour tracers, etc. range among these variables. This way, the dimensions of the equation system that must be solved for each outer (fixed-point) iteration can be reduced. Currently the computational demand roughly scales as the square of the number of components considered. Thus, even a 10% reduction in the number of components to consider for the complete system of equations would reduce the overall computational demand by about 20%. The remaining variables can then be treated without any further outer iterations once the advection rate profile has stabilised.

The numerical efficiency of MEDUSA allows for long simulation experiments required to analyse the dynamics of development of particular sedimentary features that take a long time to develop. Such features show up in steady-state simulations, but would go undetected in time-dependent simulation experiments unless run over a sufficiently long time. With interactive sediments, a biogeochemical model generally requires much longer to reach equilibrium (50 to 200 kyr). Isotope-enabled applications generally range at the upper end of the range. Vertically resolved early diagenesis models are often of similar vertical complexity (in terms of the number of vertical layers) than the biogeochemical models that they are meant to be coupled to. It is therefore important to have a model that offers the possibility of long time steps, and that can be trimmed down to the bare essentials required by any given application.

*Code availability.* The codes used in this paper are provided in the archive included in the Supplement for use under the GNU Affero General Public License, version 3 or later (the $\mu$XML library, MEDUSA, MEDUSACOCOGEN and the MEDUSA applications, including MEDMBM) or the Apache-2.0 license (LIBTHDYCT). That archive also includes the required forcing data, except for the data from the coupling simulator, which have to be downloaded from their original sources. Instructions about how to repeat all the reported simulation experiments are given in the "*Building and Running the Test Case Applications*" memo in the Supplement. The codes are furthermore archived on Zenodo (Munhoven, 2020a, b, c). Future bug-fix releases and updates will also be archived there.

*Author contributions.* GM developed the model and all of its dependencies, designed and carried out the model experiments, performed the analysis of the results, created all the graphics and wrote the paper.

*Competing interests.* The author declares that there is no conflict of interest.

*Acknowledgements.* I thank Parisa Jourabchi and Christof Meile for discussions on their working assumptions and for completing the data used in Jourabchi et al. (2008) as well as J. Keith Moore and his colleagues for making the results of their BEC simulation experiments openly available to the public. The constructive comments and suggestions from the two anonymous reviewers are greatly appreciated

and have helped to considerably improve the manuscript. Financial support for this work was provided by the Belgian Fund for Scientific Research – F.R.S.-FNRS (project SERENATA, grant CDR J.0123.19). Guy Munhoven is a Research Associate with the Belgian Fund for Scientific Research – F.R.S.-FNRS.

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

**Table 1.** General characteristics of vertically resolved sediment models used in global biogeochemical models and, for comparison, of two high-complexity models (C.CANDI and BRNS-global). The numbers of layers or nodes were taken from the respective papers and are only relevant for components whose concentration profiles are spatially resolved and not for those that are supposed to be well mixed (see Table 2). '$n$ layers' means that the number of layers is not fixed but grows as simulations proceed. For GEOCLIM *reloaded*, the number of layers was estimated from the given thicknesses of the top- and the bottom-most layers and the reported grid-point distribution function. OMEN-SED has four functional biogeochemical layers based upon redox zonation; their thicknesses adjust on the oceanic boundary conditions.

| Model and Reference | Resolution | Thickness | Core layers |
|---|---|---|---|
| Archer (1991) | 10 layers | 10 cm | — |
| Archer (1996a) | 13 layers | 10 cm | — |
| HAMOCC 2S (Heinze et al., 1999) | 10 layers | 10 cm | 1 |
| HAMOCC 5.1 (Maier-Reimer et al., 2005) | 12 layers | 14 cm | 1 |
| HAMOCC 5.2 (Ilyina et al., 2013a) | 12 layers | 14 cm | 1 |
| MUDS (Archer et al., 2002) | 17 layers | 100 cm | — |
| MEDUSA (v. 1, Munhoven, 2007) | 21 nodes | 10 cm | $n$ |
| CESM-MEDUSA (Kurahashi-Nakamura et al., 2020) | 21 nodes | 10 cm | $n$ |
| SEDGEM (Ridgwell and Hargreaves, 2007) | $1 + n$ layers | $5 + n$ cm | (included) |
| DCESS-ESM (Shaffer et al., 2008) | 7 layers | 10 cm | ? |
| GEOCLIM *reloaded* (Arndt et al., 2011) | ∼100 nodes | 100 cm | — |
| OMEN-SED (Hülse et al., 2018) | 4 layers | dynamic | — |
| C.CANDI (Luff et al., 2000) | 1000 layers | 25 cm | — |
| BRNS-global (Jourabchi et al., 2005) | 251 nodes | 165 cm | — |

**Table 2.** Porewater and solid phase species in sediment models used in global biogeochemical models and, for comparison, in two typical applications of high-complexity models (C.CANDI, which has been coupled to a regional ocean model for short-term applications of a few years only, and BRNS-global, which has been mainly used for steady-state studies of individual stations). In the solids' column, 'Clay' should be understood to stand for any inert, detrital or dilutant material.

| Model | Solutes | Solids |
|---|---|---|
| Archer (1991) | $CO_2$, $HCO_3^-$, $CO_3^{2-}$, $O_2$ | (Calcite, Clay,)[a] (OrgC)[b] |
| Archer (1996a) | $CO_2$, $HCO_3^-$, $CO_3^{2-}$, $B(OH)_3$, $B(OH)_4^-$, $O_2$ | (Calcite, Clay,)[a] OrgC |
| HAMOCC2S (Heinze et al., 1999) | $TCO_2$, TAlk, $O_2$, $PO_4^{3-}$, $Si(OH)_4$ | Clay, $^{12}$C-Calcite, $^{13}$C-Calcite, $^{14}$C-Calcite, Org$^{12}$C, Org$^{13}$C, Org$^{14}$C, Opal |
| HAMOCC5.1 (Maier-Reimer et al., 2005) | $TCO_2$, TAlk, $O_2$, $PO_4^{3-}$, $Si(OH)_4$ | Clay, Calcite, OrgC, Opal |
| HAMOCC5.2 (Ilyina et al., 2013a) | $TCO_2$, TAlk, $O_2$, $PO_4^{3-}$, Si, $NO_3^-$, Fe, $N_2$ | Clay, Calcite, OrgC, Opal |
| MUDS (Archer et al., 2000, 2002) | $CO_2$, $HCO_3^-$, $CO_3^{2-}$, $O_2$, $NO_3^-$, $Si(OH)_4$, $Mn^{2+}$, $Fe^{2+}$, $NH_4^+$ | Clay, Calcite, OrgC$_{fast}$, OrgC$_{slow}$, Opal, $MnO_2$, FeOOH |
| MEDUSA (v. 1, Munhoven, 2007) | $CO_2$, $HCO_3^-$, $CO_3^{2-}$, $O_2$ | Clay, Calcite, Aragonite, OrgC |
| CESM-MEDUSA (Kurahashi-Nakamura et al., 2020) | $CO_2$, $HCO_3^-$, $CO_3^{2-}$, $O_2$ $H_4SiO_4$, $NO_3^-$, DI$^{13}$C, DI$^{14}$C | Clay, Calcite, $^{13}$C-Calcite, $^{14}$C-Calcite, OrgC, Org$^{13}$C, Org$^{14}$C, Opal |
| SEDGEM (Ridgwell and Hargreaves, 2007) | — | Clay, Calcite, $^{13}$C-Calcite, $^{14}$C-Calcite, |
| SEDGEM (Ridgwell, 2007) | — | Clay, Calcite, $^{13}$C-Calcite, $^{14}$C-Calcite, Opal |
| DCESS-ESM (Shaffer et al., 2008) | $CO_3^{2-}$, $O_2$ | (Calcite, Clay,)[a] OrgC |
| GEOCLIM *reloaded* (Arndt et al., 2011)[c] | $TCO_2$, TAlk, THS, TB, $O_2$, $PO_4^{3-}$, $NO_3^-$, $NH_4^+$, $H_2S$, $SO_4^{2-}$, $CH_4$ | POC, PIC |
| OMEN-SED (Hülse et al., 2018)[d] | $TCO_2$, TAlk, $O_2$, $NO_3^-$, $NH_4^+$, $SO_4^{2-}$, $PO_4^{3-}$ | POC$_1$, POC$_2$ and optionally POC$_3$ |
| C.CANDI (Luff et al., 2000; Luff and Moll, 2004)[c] | $O_2$, $NO_3^-$, $Mn^{2+}$, $Fe^{2+}$, $SO_4^{2-}$, $CH_4$, $TPO_4$, $TNH_4$, $H_2S$, $HS^-$, $CO_2$, $HCO_3^-$, $CO_3^{2-}$ | $MnO_2$, $Fe(OH)_3$, POC$_{\#0}$, POC$_{\#1}$, POC$_{\#2}$, FeS |
| BRNS-global (Jourabchi et al., 2005)[c] | $CO_2$, $HCO_3^-$, $CO_3^{2-}$, $O_2$, $NO_3^-$, $Mn^{2+}$, $Fe^{2+}$, $NH_4^+$, $Ca^{2+}$, $SO_4^{2-}$, $H_2S$, $HS^-$, $CH_4$, $B(OH)_3$, $B(OH)_4^-$ | OrgC, Calcite, $MnO_2$, $Fe(OH)_3$, FeS, $FeCO_3$, $MnCO_3$ |

[a] Supposed to be well mixed, i. e., only total contents of the bioturbated layer traced.

[b] OrgC concentration profile prescribed following Emerson (1985).

[c] Solids' advection rate profile prescribed and therefore no clay or other inert solid component considered.

[d] Solids' burial rate prescribed and therefore no clay or other inert solid component considered.

**Table 3.** Reaction rate law expressions and parameter values used for the MBM experiments. $C_{cs}$ is the $CO_3^{2-}$ concentration in seawater at saturation with respect to calcite and $C_{as}$ that for aragonite; $C_{hox}$ is the half-saturation concentration of $O_2$ for the oxic remineralisation of organic matter. $(\ldots)^+$ denotes the positive part of $(\ldots)$. The different reaction rates, $\hat{R}$, are expressed in terms of the dissolving/remineralised solid, in $kg\,(m^3\,total\,sediment)^{-1}$. Concentrations of solids are expressed in $kg\,(m^3\,solid\,sediment)^{-1}$, those of solutes in $mol\,(m^3\,porewater)^{-1}$.

| | |
|---|---|
| Calcite dissolution | $\hat{R}_{cdis} = k_c\,(1-\varphi)\,[\text{Calcite}]\,((1-[CO_3^{2-}]/C_{cs})^+)^{4.5}$ |
| | with $k_c = 365.25\,\text{yr}^{-1}\,(= 100\%\,\text{day}^{-1})$, $C_{cs} = C_{cs}(S,T,p)$ |
| Oxic OM remineralisation | $\hat{R}_{omox} = k_{ox}\,(1-\varphi)\,[\text{OrgM}]\,([O_2]/(C_{hox}+[O_2]))$ |
| | with $k_{ox} = 0.024\,\text{yr}^{-1}$, $C_{hox} = 20\,\mu\text{mol}\,L^{-1}$ |
| Aragonite dissolution | $\hat{R}_{adis} = k_a\,(1-\varphi)\,[\text{Aragonite}]\,(C_{as}-[CO_3^{2-}])^{1.87}$ |
| | with $k_a = 53.8\,\text{yr}^{-1}\,(\text{mol}^{-1}\,\text{m}^3)^{1.87}$, $C_{as} = C_{as}(S,T,p)$ |

**Table 4.** Reaction rate law expressions adopted for COUPSIM and parameter values for the experiment depicted on Fig. 4 (represented by the full symbols on Fig. 3). Units and notations are the same as in Table 3. In addition, $C_{\mathrm{hnr}}$ is the half-saturation concentration of $NO_3^-$ and $C_{\mathrm{io}}$ the characteristic inhibition concentration of $O_2$ for the oxidation of organic matter by nitrate reduction.

| | |
|---|---|
| Calcite dissolution | $\hat{R}_{\mathrm{cdis}} = k_{\mathrm{c}}\,(1-\varphi)\,[\mathrm{Calcite}]\,((1-[CO_3^{2-}]/C_{\mathrm{cs}})^+)^{4.5}$ |
| | with $k_{\mathrm{c}} = 36.525\,\mathrm{yr}^{-1}\ (= 10\,\%\,\mathrm{day}^{-1}),\ C_{\mathrm{cs}} = C_{\mathrm{cs}}(S,T,p)$ |
| Oxic OM remineralisation | $\hat{R}_{\mathrm{omox}} = k_{\mathrm{ox}}\,(1-\varphi)\,[\mathrm{OrgM}]\,([O_2]/(C_{\mathrm{hox}} + [O_2]))$ |
| | with $k_{\mathrm{ox}} = 0.015\,\mathrm{yr}^{-1},\ C_{\mathrm{hox}} = 3\,\mathrm{\mu mol\,L}^{-1}$ |
| OM remin. by nitrate reduction | $\hat{R}_{\mathrm{omnr}} = k_{\mathrm{nr}}\,(1-\varphi)\,[\mathrm{OrgM}]\,([NO_3^-]/(C_{\mathrm{hnr}} + [NO_3^-]))\,(C_{\mathrm{io}}/(C_{\mathrm{io}} + [O_2]))$ |
| | with $k_{\mathrm{nr}} = 0.005\,\mathrm{yr}^{-1},\ C_{\mathrm{hn}} = 30\,\mathrm{\mu mol\,L}^{-1},\ C_{\mathrm{io}} = 10\,\mathrm{\mu mol\,L}^{-1}$ |
| Opal dissolution | $\hat{R}_{\mathrm{odis}} = k_{\mathrm{o}}\,(1-\varphi)\,[\mathrm{Opal}]\,(C_{\mathrm{os}} - [H_4SiO_4])^+$ |
| | with $k_{\mathrm{o}} = 0.05\,\mathrm{yr}^{-1}\,\mathrm{mol}^{-1}\,\mathrm{m}^3,\ C_{\mathrm{os}} = 700\,\mathrm{\mu mol\,L}^{-1}$ |

**Table 5.** Execution times of the COUPSIM best-fit experiment. One unit of CPU time corresponds here to 9:14.73 min. The executing platform had an Ubuntu 16.04.6 LTS operating system (64-bit kernel 4.4.0-185-generic) running on a 1.90 GHz Intel® Core™ i7-8650U CPU; the codes were compiled with GFORTRAN 5.4.0, using optimisation level -O0. Experiments carried out with the MPI interfaces used OPENMPI v. 1.10.2 and were run on as many processes as indicated in brackets.

| Number of steps | Step length | Wall-clock time (relative units) | | | | |
|---|---|---|---|---|---|---|
| | | Serial | MPI-1D (2) | MPI-2D (2) | MPI-1D (4) | MPI-2D (4) |
| 1 | $\infty$ | 0.049 | 0.034 | 0.028 | 0.021 | 0.020 |
| 1 | 1000 yr | 0.028 | 0.020 | 0.016 | 0.012 | 0.010 |
| 10 | 100 yr | 0.17 | 0.12 | 0.097 | 0.079 | 0.065 |
| 100 | 10 yr | 1.0 | 0.73 | 0.62 | 0.47 | 0.42 |
| 1000 | 1 yr | 7.7 | 5.9 | 4.8 | 3.5 | 3.2 |

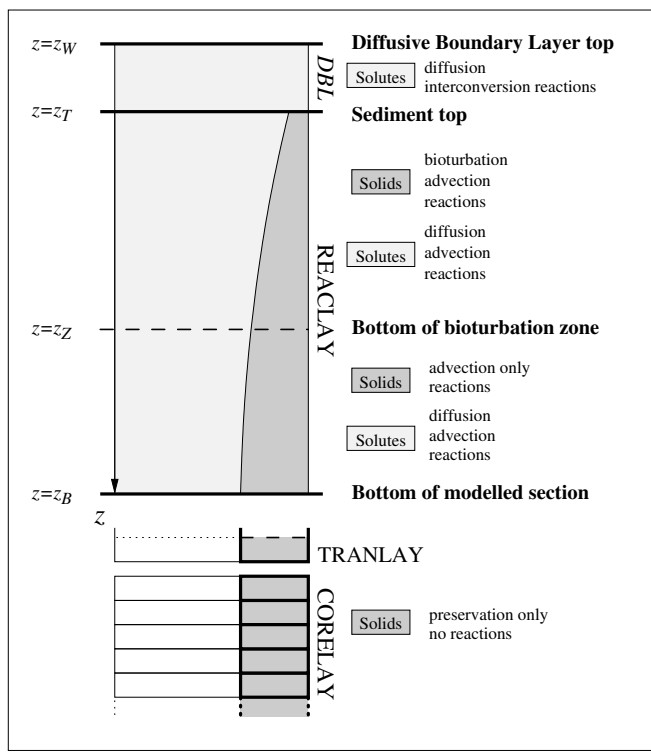

**Figure 1.** Partitioning of the sediment column in MEDUSA: an optional diffusive boundary layer (DBL) on top of the main part of the model sediment where diagenetic reactions and advective-diffusive transport take place (REACLAY), the transition layer (TRANLAY) and the core represented by the stack of layers (CORELAY). The bottom of the bioturbation zone may coincide with the bottom of REACLAY. If the optional DBL is omitted, $z_W = z_T$. See text for further details.

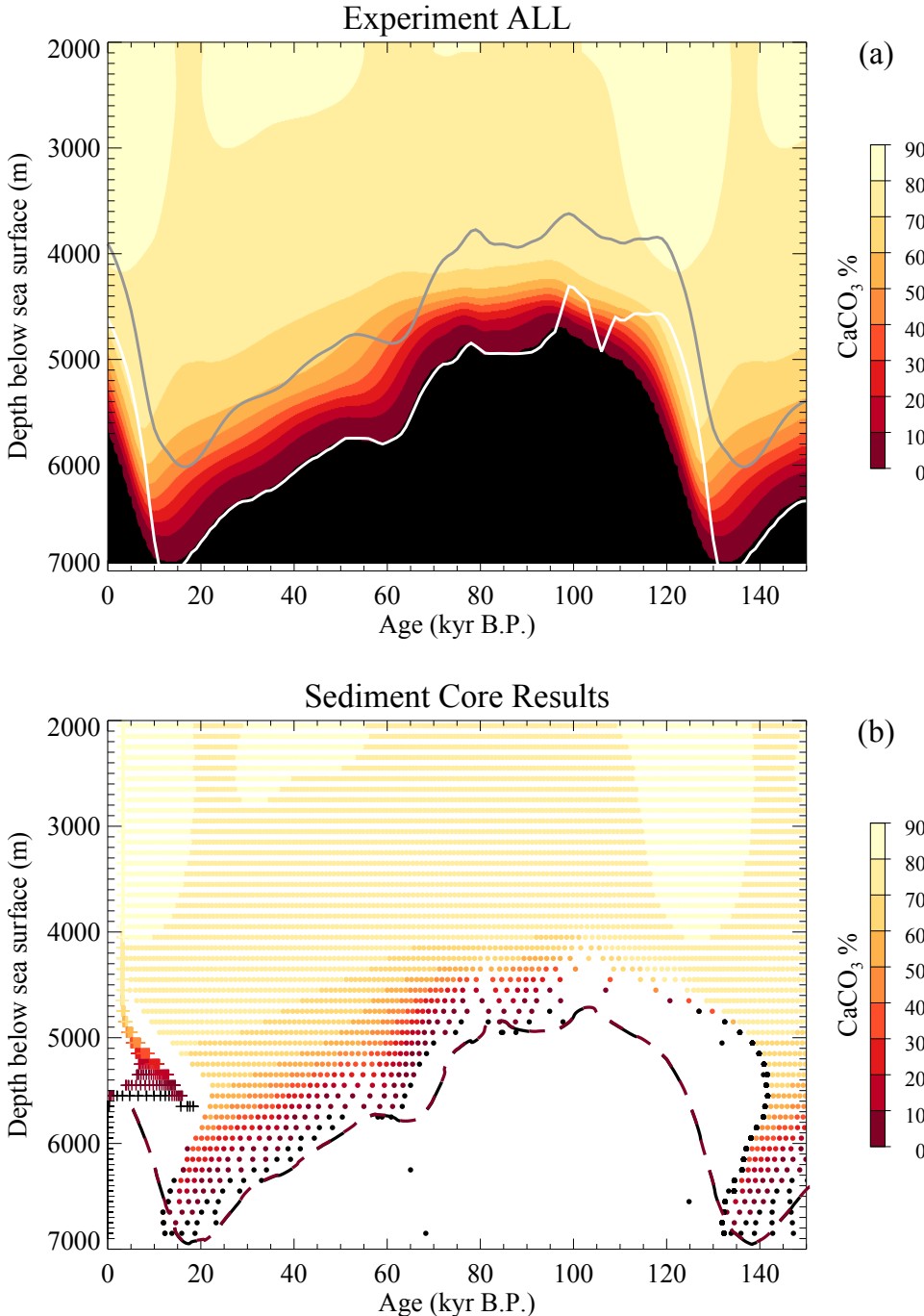

**Figure 2.** Average surface sediment CaCO₃ content for the ALL experiment from Munhoven (2007). (a) Actual surface sediment %CaCO₃ as a function of time. The grey line traces the evolution of the calcite saturation horizon (CSH), the white line the carbonate compensation depth (CCD) where the dissolution rate equals the deposition rate. (b) Synthetic sediment core record from cores "drilled" at 100 m sea-floor depth intervals from 2050 to 6850 m d.b.s.l. in the low-latitude Indo-Pacific box, as a function of age of the layers. The long dashed line indicates the limit between the black and maroon zones in panel (a), shifted by 5 kyr to the right.

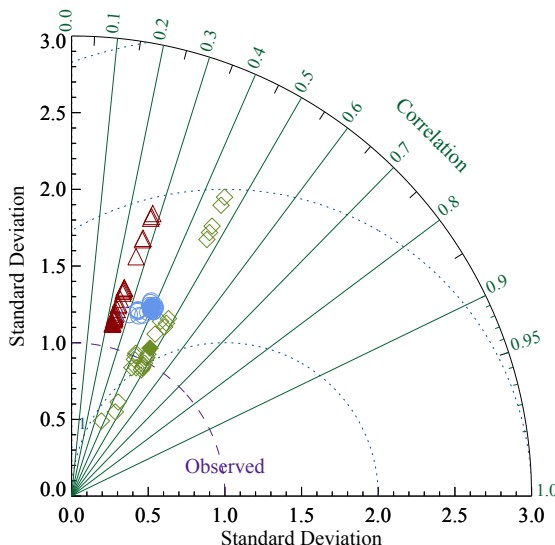

**Figure 3.** Taylor diagram of the results of the parameter sensitivity experiments carried out with COUPSIM: %Calcite (blue circles); %TOC (green diamonds); %Opal (red). The full symbols indicate the combination found for the best-fit experiment. Maps for the surface sediment compositions of this best-fit experiment are shown on Fig. 4.

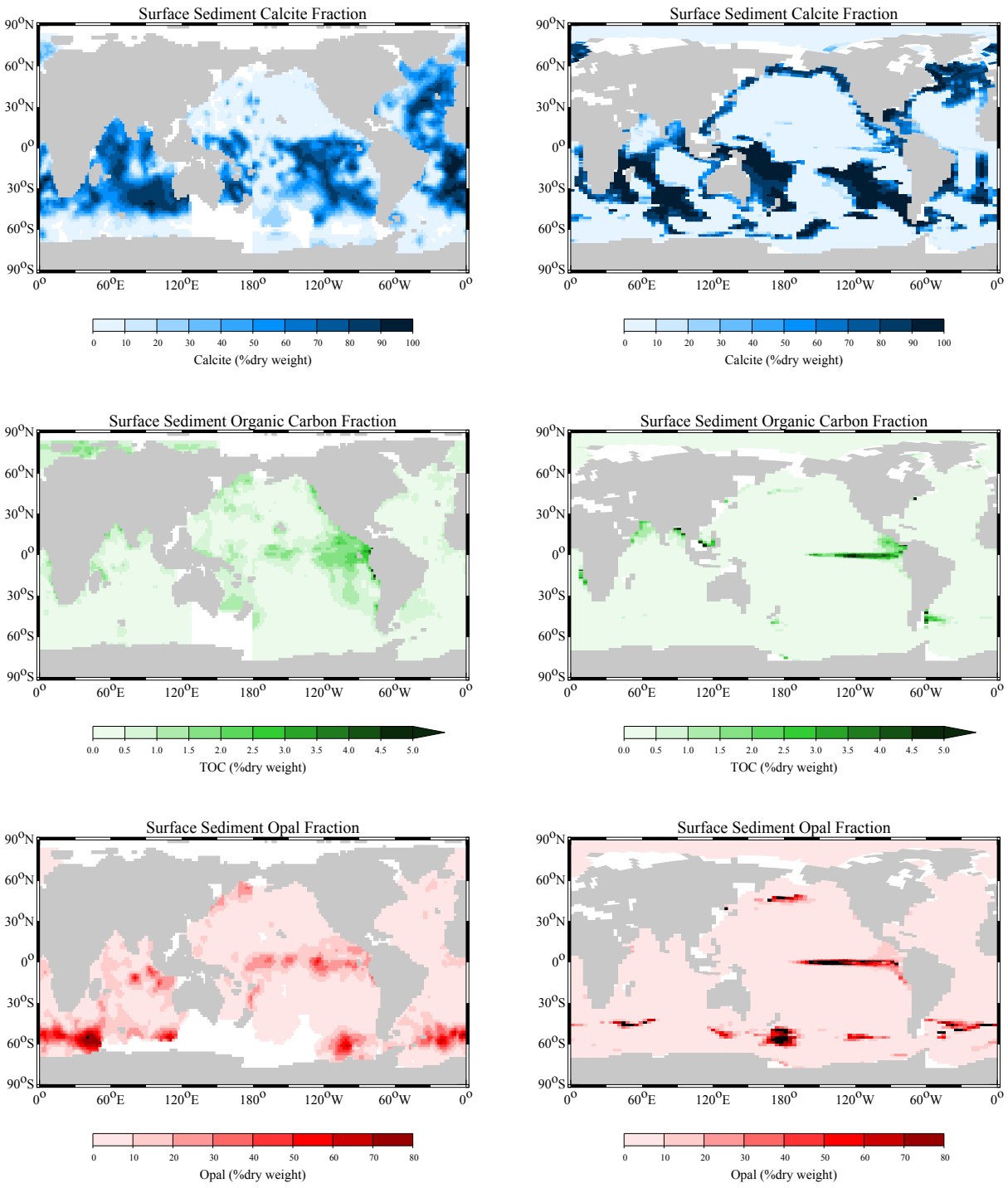

**Figure 4.** Left column: data (Seiter et al., 2004), binned to a $2° \times 2°$ resolution comparable to the average resolution of BEC; right column: COUPSIM (i.e., MEDUSA forced by the BEC model results of Moore et al. (2004)).

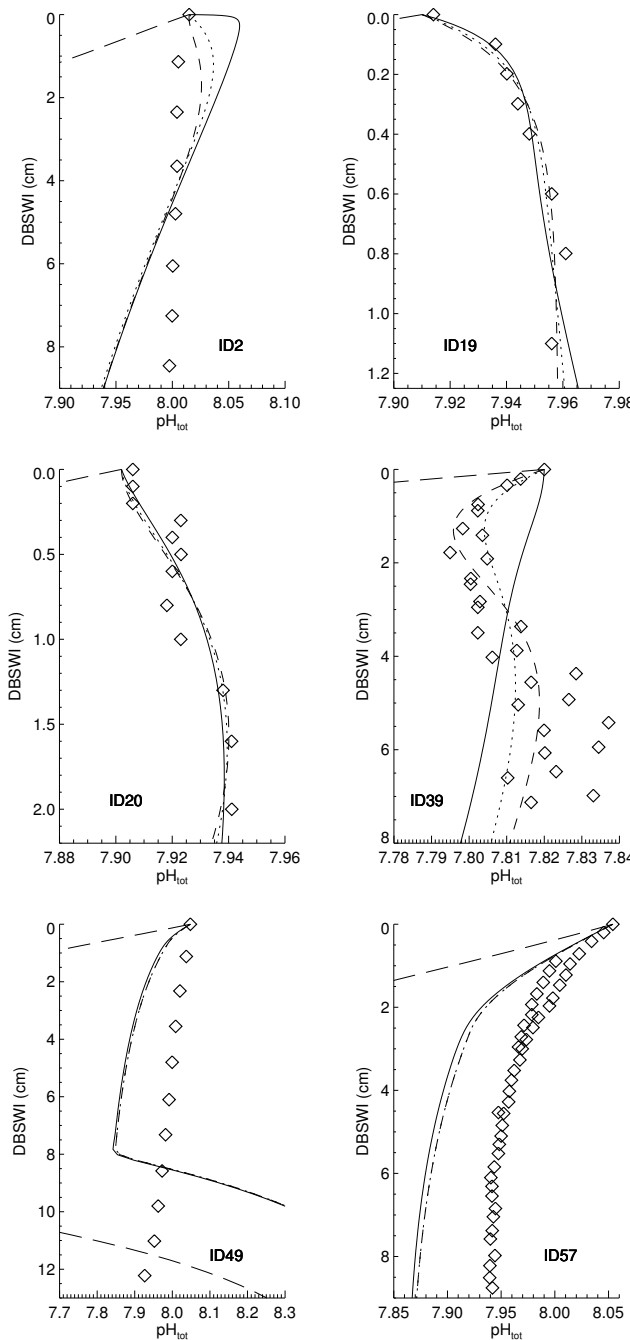

**Figure 5.** $p$H profiles, unconstrained with respect to the calcite mass fraction in the deposition flux and in the surface solids, for stations that produced physically impossible fits (calcite mass fractions exceeding 100% either in the mass deposition flux or the solids concentration profile). Lines are as follows: long dashed – no calcite dissolution; full – calcite dissolution kinetics of order 4.5; dotted – order 2; short dashes – linear.

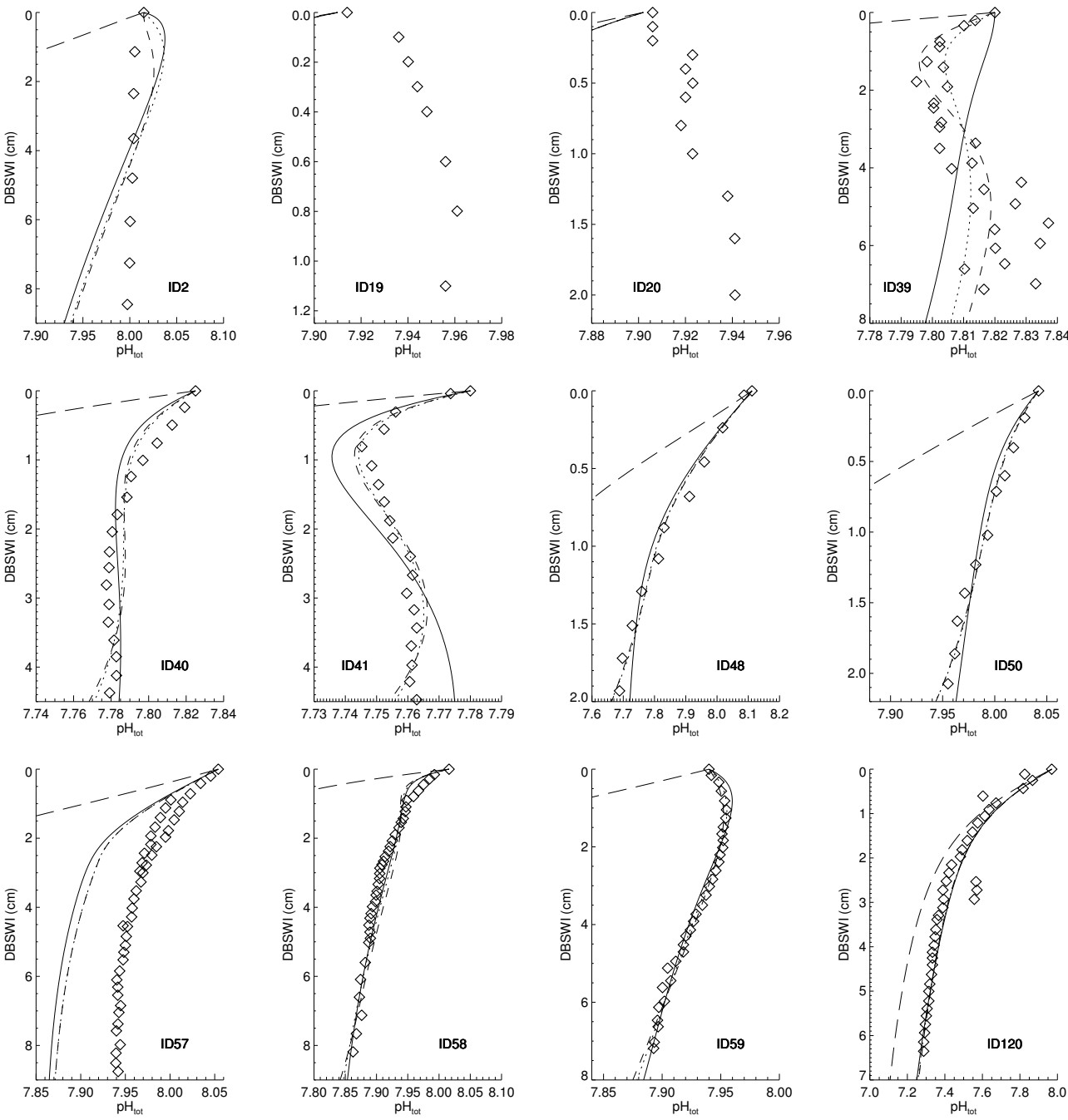

**Figure 6.** $p$H profiles for experiments with calcite mass fractions in the deposition flux limited to 90%. Lines are as follows: long dashed – no calcite dissolution; full – calcite dissolution kinetics of order 4.5; dotted – order 2; short dashes – linear.

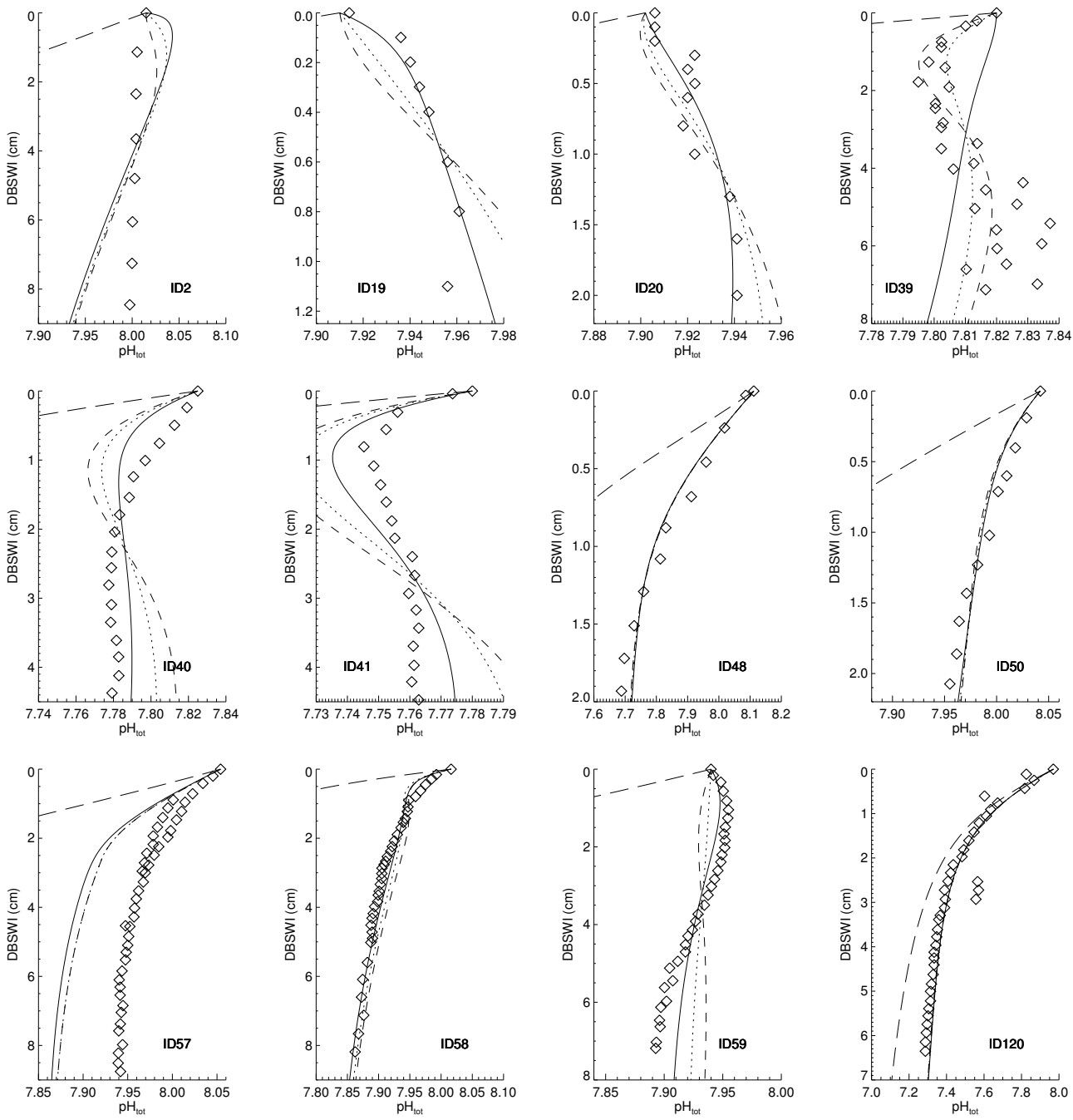

**Figure 7.** *p*H profiles with data-constrained surface sediment %calcite. Lines are as follows: long dashed – no calcite dissolution; full – calcite dissolution kinetics of order 4.5; dotted – order 2; short dashes – order 1 (linear). Please notice that the $n = 4.5$ fit at site 2 and all the fits for sites 19 and 20 require calcite mass fractions in the deposition fluxes exceeding 100% (between 140 and 730%).