# Peer review of "Model of Early Diagenesis in the Upper Sediment with Adaptable complexity – MEDUSA (v. 2): a time-dependent biogeochemical sediment module for Earth System Models, process analysis and teaching"

_Geoscientific Model Development, 2020_

## Referee Comment (RC1) · Anonymous Referee #1 · 8 Nov 2020

Guy Munhoven presents a revised version of his time-dependent vertically resolved biogeochemical model of early diagenesis (MEDUSA) described in an excellent model development paper. In particular the model code has been thoroughly revised to create a more flexible simulation environment for reaction transport calculations that can be specified by the user and thus easily tailored for specific applications. A code generator has been developed that produces the respective parts of the source code and a complete reference guide is provided as a supplement to the manuscript. The new model

code facilitates the coupling to common grid layouts of ocean biogeochemical models and the coupling procedure is explained and exemplified in additional supplementary documents. The numerical efficiency of MEDUSA-v2 allows for its coupling to Earth system models and the execution time can be further reduced via parallel processing of multiple sediment-columns.

Overall, the revised model represents a very useful and important tool for a multitude of applications. The model description is very thorough and precise, including many excellent documents as a supplement. I only have a few minor comments about the description of the solution strategy and a few technical comments the author should address before publication.

2.2.2 Solution strategy

The different initial conditions for the iteration scheme are excellent and clear. However, I think it would be good if the convergence criteria is made more clear in the main manuscript. From your "Technical Reference" I understand that you test for a convergence of the (solid) concentration profiles. The second criterion, however, is difficult to comprehend. I think it would be good to include a clear description of both criteria in the main manuscript. Also, is the overall solution divided into different steps? E.g. is OM degradation and the resulting profile calculated first and from it MEDUSA deduces the O2 profile? Because the zonation of oxic and anoxic OM degradation has implications for the production of alkalinity and thus carbonate dissolution. Finally, does MEDUSA check for the convergence of burial velocity at the very end if the solid components are not declared as volumeless?

Specific/technical comments:

page 1: ln. 21: I think it should read "the latter" here

Page 2: ln. 25/26: Maybe also add: …or organic matter completely oxidized even if oxygen levels are low and/or organic matter input is very high.

ln. 46: Please give example reference for the meta-model approach.

ln. 49 - 50: Please add references for examples for the " expert-chosen empirical parametric functions" and for the " system identification theory" approach

Page 3: Table 1+2 and in the corresponding text (e.g. ln 70 and first paragraph on pg. 5): For completeness please include OMEN-SED (Hülse et al., (2018) which is an analytical early diagenetic steady-state model for OM degradation dynamics with flexible resolution. OMEN-SED is available as stand-alone model and coupled to cGENIE and presents a noval approach to simulate benthic-pelagic coupling (i.e. different to the approaches presented in Table 1 and 2).

Page 5: Ln 127: ". . . during which some parts gets remineralised (i.e., oxidised or dissolved), and some parts gets buried." Please check the spelling here.

Ln 130-133: Please add reference.

ln 134: " Previously buried carbonates will then return to the sedimentary mixed as a result of the bioturbation activity..." I guess the "sedimentary mixing" is a typo, please rephrase.

Page 7: ln 142: I think the "instead of" can be deleted

Page 9:

ln 197: So for solutes DBL is the same as REACLAY only that porosity equals 1? Could you please include a brief explanation why/when a DBL is important and when it should be used?

Page 10: ln 215: Pointing the reader here to you supplementary document "Early Diagenesis in Sediments – A one-dimensional model formulation" would be good as it provides more and very useful information for instance about the parameterisations for tortuosity and how the diffusion coefficients are calculated. Maybe also cite Ullman and Aller (1982) who did a lot of early work on diffusion coefficients and tortuosity

parameterization.

ln. 221: bioirrigation "constant". What are you using for alpha in the set-up used in your simulations for this manuscript? Is it a depth dependent approach (e.g. Soetaert et al. (1996))?

Page 11: Additional constraints: Maybe it makes sense to include the constraint that the porosity profile is prescribed (i.e. time independent, $d\varphi/dt = 0$) at this point?

Page 12: ln. 292: "... and the topmost interior node of REACLAY" Is this part of the sentence not redundant as this note is always below the top of REACLAY? The same applies for the bottom of DBL.

Ln 297: " but the spacing and extent of each of these may be different." Depending on what? E.g. if the user wants to have a higher resolution for shelf-sediments compared to the deep ocean?

Page 16: ln 405: I think this should read: dissolved organic matter instead of "dissolved inorganic matter"

Page 17: ln 438: I think it should read: simulation experiments from Munhoven (2007)

ln 450-451: I guess the sentence could be changed to: "MBM is an eleven-box carbon cycle model of the carbon cycle in the ocean and the atmosphere."

ln 453: close the parenthesis after Pacific

Section 3.2 COUPSIM – Coupling simulator page 21 ln. 519: why did you just simulate depths greater than 1000m below sea-level? And are the shallower observations in Seiter (where generally the highest TOC concentrations are observed) excluded from the data-set (or is there hardly any data for shallower settings because of the resolution of the observations)?

Fig. 3 : I think opal are the red triangles. In that case the caption needs to be corrected.

Ln 551: Your rate constants for OM degradation could be for instance compared to Palastanga et al. (2011) who used the HAMOCC model coupled to a diagenetic model using a 1G-approach for OM. They also compare their results to the Seiter et al. (2005) data and find the best fit using k_ox=0.005 yr-1 and k_anox=0.002 yr-1 for depths > 2000m (k_ox=0.01 yr-1 and k_anox=0.008 yr-1 for depth < 2000m).

page 27 ln 652: spelling: I think it should read: ". . . in order to allow..."
* * *

---

## Referee Comment (RC2) · Anonymous Referee #2 · 1 Mar 2021

Review of 'Model of Early Diagenesis in the Upper Sediment with Adaptable complexity – MEDUSA (v. 2): a time-dependent biogeochemical sediment module for Earth System Models, process analysis and teaching' by G. Munhoven

General comments

The author presents a description of a flexible novel 1D reactive transport model of early diagenesis MEDUSA in version 2. The description is quite concise and easy to read. Focus of the model description is rather on the coupling to global biogeochemical cycling models. The model and its documentation will be useful to the community. Publication in GMD would be most appropriate and hence I recommend the publication. I have only minor comments to ask the author for clearer descriptions regarding model feature and application.

(1) The model seems to apply different numerical schemes to different realms as inferred from Fig. 1. However, how the model does this is not 100% clear from the ms. Also, there seems to be some options that could affect the overall solution scheme/sequence, e.g., whether the model includes diffusion boundary layer or not; whether the advection is solved or not; whether the model tracks time or not, etc. Although the details are referred to the Supplement when relevant, a flow chart of the calculation scheme/sequence including branches for some important model options would be very helpful for the reader to grasp what is going on in the model.

(2) Pros and cons of options are discussed but it is not 100% obvious to the reader when to adopt which option. For example, in the ALL experiment, the author discussed the difference between the tracked time and actual time and the cause of it, but not sure when/why we should use the tracked time. Also, the author described that the implementation of mineral volume options (related to advection scheme) does not affect the overall results, which will make the reader to wonder when/why to adopt which option regarding mineral volumes. Furthermore, under what conditions should we adopt the option of DBL? As a model description paper, providing a conclusion on the options may not be obligation, but guiding the reader a little bit more could be desired.

Specific comments

Table 1: In my understanding $CaCO_3$ and clay are not vertically resolved in Archer's original model (1991); only porewater chemistry and OM are vertically resolved. How do you define the layer number when layer numbers are different between different species as is the case for Archer (1991)?

Line 248: Are you saying that one of solid species is treated as a dilatant material and does not necessarily complies with advection law (e.g., Eq. (1) + equation in line 205)? I thought iterative

implementation of Eqs. (1) and (6) and equation in line 205 could satisfy Eq. (4) (e.g., Archer et al., 2002).

Lines 258-270: Not sure whether the equations for different realms are solved at once or in a sequence with/without iterations. Is it appropriate to define a boundary condition for TRANSLAY as done for the above layers?

Line 327: Do you mean $w$ is calculated time-explicitly but solids-solutes are calculated time-implicitly? Also, I suppose Newton iteration is conducted only in REACLAY and DBL? Do you separate calculations between REACLAY and DBL or at once?

Line 499: Is MEDUSA fully coupled to BEC (allowing exchange fluxes passed between the two models/modules) or are you just using the BEC output as boundary conditions and not returning any fluxes to BEC? The steady-state results of sedimentary profiles between the above two cases could different.

Table 3, cap. L1: Isn't 1L = 1dm$^3$ universally correct? If so this does not have to be assumed?

Technical comments

Line 134: 'the sedimentary mixed' should be replaced with 'the sedimentary mixed layer'?

Line 453: Right parenthesis is missing.

Tables 1 and 2: 'BRNS-global' or 'BRNS-GLOBAL'?

Fig. 3 caption L2: %Opal should be red triangles?

---

## Author Comment (AC1) · 17 Mar 2021

I thank the referee for their careful and in-depth reading of my manuscript and their constructive comments. For the sake of conciseness, I reply here only to questions and comments where extra information was requested. I gladly accept all the other suggestions and corrections and will take them into account for the revision of the

manuscript.

**General comments**

*2.2.2 Solution strategy*

*The different initial conditions for the iteration scheme are excellent and clear. However, I think it would be good if the convergence criteria is made more clear in the main manuscript. From your "Technical Reference" I understand that you test for a convergence of the (solid) concentration profiles. The second criterion, however, is difficult to comprehend. I think it would be good to include a clear description of both criteria in the main manuscript. Also, is the overall solution divided into different steps? E.g. is OM degradation and the resulting profile calculated first and from it MEDUSA deduces the O2 profile? Because the zonation of oxic and anoxic OM degradation has implications for the production of alkalinity and thus carbonate dissolution. Finally, does MEDUSA check for the convergence of burial velocity at the very end if the solid components are not declared as volumeless?*

The adopted **convergence criteria** are standard practice in the numerical solution of non-linear equations and equation systems (e.g., Antia, 2002). When iteratively solving an equation or equation system, there are essentially two main types of iteration stopping criteria that can be adopted: one based upon the differences between subsequent iterates and one based upon the magnitude of the function residual. None of these is universally adequate and it is generally advisable to use a combination of the two. For safety reasons, they should furthermore be complemented by a limitation of the total number of iterations (Kincaid and Cheney, 1991). With MEDUSA, I found that monitoring the difference between subsequent iterates and stopping iterations once this difference fell below a given (small) threshold value

would not always provide an acceptable solution to the equation system. In some instances, where iterations were only slowly progressing at a high function residual value, iterations would often be stopped prematurely on the basis of that criterion alone and convergence falsely diagnosed. We therefore first of all require that the function residual value is sufficiently small. "Sufficiently small" clearly makes sense only if the scales of the equations can be properly estimated. Once the scaled function residual is sufficiently small (first criterion), it is interesting to perform some root refinement (**second criterion**). Here, this is only done if the total number of iterations has not yet been exceeded. In general, the second criterion requires a few extra iterations only and may further reduce the function residual by several orders of magnitude. As suggested by the referee, this will be made more explicit in the revised manuscript.

Regarding the actual implementation of the convergence test, the *Technical Reference* is possibly misleading in its current form as the scaling of the equations is only detailed for solids therein. Similar scaling is nevertheless applied for solutes and it is straightforward to derive the scaled expressions of the solutes' equations from those of the solids. It is sufficient to notice that solute diffusion is an intraphase type diffusion, and $w$ may be substituted for by $u$. The convergence test considers all the scaled equations for all the solids and all the solutes (criterion 1) and all the scaled increments of all the solids' and solutes' concentrations (criterion 2). This will be made more explicit in the *Technical Reference*, by also providing detailed expressions for solutes.

The **solution strategy** is an all-at-once one – no separate process-based steps. As the framework system must be able to cope with any composition and reaction network, all processes are treated on par, and except for positivity of the concentrations, no further a priori assumptions are made.

If organic matter is included as a component and other oxidation pathways than the oxic one are considered, the redox zonation emerges automatically in the solution of the equation system, once that solution becomes sufficiently precise.

The **convergence of the solids' advection rate profile** is currently not explicitly checked for. It is expected that constituent concentrations keep changing as long as the advection rate profile changes from one iteration to another. It is therefore unlikely that iterations get stopped unless the advection rate profile stabilises. For extra security, it might nevertheless be interesting to include a test for the convergence of the advection rate profile as well. So far, this has, however, not been necessary.

**Specific/technical comments**

*ln. 46: Please give example reference for the meta-model approach.*

References to Sigman et al. (1998) and Dunne et al. (2007) will be added.

*ln. 49 - 50: Please add references for examples for the "expert-chosen empirical parametric functions" and for the "system identification theory" approach*

For the "expert-chosen empirical parametric functions," reference will be made to Dunne et al. (2007). The "system identification theory" approach was actually meant as a future perspective. It has to my best knowledge so far only been used at a more global scale for ESMs, but not yet to sediment models. A reference to Crucifix (2012) will be added for the fundamentals

of this approach in that more general context and to Ermakov et al. (2013) for an pilot application of this technique to MBM-MEDUSA.

*Ln 130-133: Please add reference.*

Added reference to Archer et al. (1998).

*ln 134: "Previously buried carbonates will then return to the sedimentary mixed as a result of the bioturbation activity..." I guess the "sedimentary mixing" is a typo, please rephrase.*

"the sedimentary mixed" should actually have read "the sedimentary mixed layer" — corrected.

*Page 9: ln 197: So for solutes DBL is the same as REACLAY only that porosity equals 1? Could you please include a brief explanation why/when a DBL is important and when it should be used?*

Yes, exactly: solutes are treated the same way in the DBL and in REACLAY, with porosity set to 1 in the DBL. To keep the structure of the equation system simple, equations for solids are nevertheless included in the DBL, but their concentrations are forced to zero there. The DBL acts as a transport barrier for the exchange of solutes between the sediment and the overlying seawater across the sediment-water interface: the thicker it is, the stronger the resistance it exerts. The existence of a DBL is merely due to the presence of a sediment-water interface that delimits a turbulent medium. As such, it should actually be included in any model configuration a priori.

However, including a DBL slows down a model. An important question is then to what extent the model results are influenced by the presence or absence of a DBL. I am currently writing up a manuscript on that subject and it seems that there is no unequivocal response to that question. The presence of a DBL turns out to play an important role with reduced complexity model compositions (such as the classical calcite–clay–$CO_3$ composition of Keir (1982), recently used by Sulpis et al. (2018)). With sediment compositions commonly used in sediment modules included in ocean biogeochemical models (with calcite, clay and organic matter as solids and $CO_2$, $HCO_3^-$, $CO_3^{2-}$, $O_2$ and possibly $NO_3^-$ as solutes, the role of a DBL, even of 1 mm thickness, on the preservation patterns of sedimentary material is very limited. In other configurations, it may nevertheless be important again. Sulpis et al. (2018) have calculated the global distribution of the DBL thickness over the seafloor. They find a wide range of values, from 100 to 10,000 $\mu$m, with a most probable value close to 1000 $\mu$m though.

Interestingly, early instances of simple carbonate diagenesis models generally included a DBL (e.g., Schink and Guinasso Jr., 1977). Later, DBLs "vanished" from such models (e.g., Keir, 1982) without any explicit reason. This evolution might, however, possibly be linked to the progressive adoption of non-linear calcite dissolution kinetics in such models around that time: with the linear kinetics, they can be solved analytically with or without a DBL; with non-linear kinetics, a DBL prevents such an analytical solution and even without a DBL, they only allow an approximative analytical solution. In the later developed more complex early diagenesis models for which numerical solution procedures are adopted that can just as well handle configurations with and without DBLs, DBLs are still not widespread: on one hand, CANDI (Boudreau, 1996) and OMEXDIA (Soetaert et al., 1996) include one; on the other hand Archer (1991, 1996), Dhakar and Burdige (1996), STEADYSED1 (Van Cappellen and Wang, 1996) MUDS (Archer et al.,

2002) or Jourabchi et al. (2005) do not include one.

The equation system becomes to some extent "cleaner" at the sediment water interface, SWI, once a DBL is included. Without a DBL the SWI lies on a grid node, which is ideal for a prescribed concentration boundary condition (used for solutes) but less so for a flux boundary condition (used for solids), although the discretized equations remain mathematically consistent, which is not the case when concentrations have to be prescribed at a grid vertex (Hundsdorfer and Verwer, 2003). With a DBL, the SWI lies on a grid vertex, which is then perfect for both solids and solutes, as they have to fulfil flux continuity equations in this case.

Bottom line: I recommend to include a DBL. For site-related applications, this nevertheless requires the knowledge of the thickness; for global applications, an average value of 1 mm should be adequate in most situations. For backwards compatibility with the interfaces to several biogeochemical cycle models that MEDUSA has been coupled to and that link to the MEDUSA's SUBVERSION repository for this purpose, the default in the code will nevertheless remain "no DBL" for the time being.

The revised manuscript will include a new section about the importance of a DBL and some other options that MEDUSA offers (see also comments by Anonymous Referee #2)

*Page 10: ln 215: Pointing the reader here to you supplementary document "Early Diagenesis in Sediments "A one-dimensional model formulation" would be good as it provides more and very useful information for instance about the parameterisations for tortuosity and how the diffusion coefficients are calculated. Maybe also cite Ullman and Aller (1982) who did a lot of early work on diffusion coefficients and tortuosity parameterization.*

Thank you for this suggestion. I will refer to that document and include a

reference to Ullman and Aller (1982) therein.

*ln. 221: bioirrigation "constant". What are you using for alpha in the set-up used in your simulations for this manuscript? Is it a depth dependent approach (e.g. Soetaert et al. (1996))?*

The simulation experiments presented in the manuscript use $\alpha \equiv 0$. The original MBM-MEDUSA did not include biorrigation and neither did Jourabchi et al. (2008) upon which JEASIM was based; COUPSIM is mostly focusing on great depths, where bioirrigation is negligible (Glud, 2008). As mentioned on the line 222, the bioirrigation "constant" $\alpha$ may be depth-dependent. It may even be time-dependent and it may also depend on other parameters, such as the supply rate of organic matter or other constituent concentrations. Its value is updated at each iteration at the same time as the solids' advection rate profile.

*Page 11: Additional constraints: Maybe it makes sense to include the constraint that the porosity profile is prescribed (i.e. time independent, $d\varphi/dt = 0$) at this point?*

The porosity profile is not a constraint in the same sense as the volume conservation equation. While the basic set of equations used in the model could be adapted to deal with time-dependent porosity (either prescribed or composition dependent, such as in Zeebe and Zachos (2007)), the volume conservation equation will always remain valid. I thus keep this part unchanged.

*Page 12: ln. 292: "...and the topmost interior node of REACLAY" Is this part of the sentence not redundant as this note is always below the top of REACLAY? The same applies for the bottom of DBL.*

This observation is indeed correct. I nevertheless prefer to keep the current text as it is to remind that the grid node distribution on `REACLAY` is different in the model configurations with and without a `DBL`.

*Ln 297: "but the spacing and extent of each of these may be different." Depending on what? E.g. if the user wants to have a higher resolution for shelf-sediments compared to the deep ocean?*

Yes exactly: spacing and extent can be changed depending on the users' needs, and one typical usage of this facility would be different grid-point distributions at shallow and at great depths.

*Page 17: ln 438: I think it should read: simulation experiments from Munhoven (2007)*

No, the text is correct. Munhoven (2007) included three different experiments: CRR, BST and ALL. Here, only the ALL experiment is revisited.

*ln 450-451: I guess the sentence could be changed to: "MBM is an eleven-box carbon cycle model of the carbon cycle in the ocean and the atmosphere"*

This will be changed to "`MBM` is an eleven-box model of the carbon cycle in the ocean and the atmosphere"

*Section 3.2 COUPSIM — Coupling simulator page 21 ln. 519: why did you just simulate depths greater than 1000m below sea-level? And are the shallower observations in Seiter (where generally the highest TOC concentrations are observed) excluded from the data-set (or is there hardly any data for shallower settings because of the resolution of the observations)?*

Here we limit ourselves to depths of 1000 m and greater for the simulation experiment because the input data read in from the `BEC` results lead to even more unrealistic results at shallower depths. The model works without any problems for depths shallower than 1000 m as can be easily verified by setting `dlimzw_top` in the `&nml_regridding` namelist in the configuration file `medusa_coupsim_setup.nml` in `medmbm/work/coupsim_bec` to a value lower than `1000.0D+00`. However, with shallower threshold depths one ends up with TOC contents of 40% and more (equivalent to about 90% of organic matter) in the Kara Sea, the Sea of Okhotsk and in the South China Sea, far above the highest observed value (15% in one single grid point). The 1-G approach may not be adequate here and the adopted redox process network not sufficiently complete. The model complexity for `COUPSIM` was chosen to match the available boundary conditions as far as possible.

The complete TOC dataset of Seiter et al. (2004) actually includes 36,236 data points on a $1° \times 1°$ longitude-latitude grid (64,800 grid points). Only 4 of these had TOC contents greater than 10%; about 30 of them present TOC contents between 5 and 10%. For the model-data comparison, only data for seafloor depths greater than 1000 m have been included on the maps in Figure 4, to be consistent with the seafloor extent considered.

---

## Author Comment (AC2) · 17 Mar 2021

I thank the referee for their careful consideration of my manuscript and for pointing out several shortcomings in the text. Please find below my replies to those comments that asked for information or clarification. All the other suggestions and corrections not mentioned here will be integrated in the revised text as recommended.

**General comments**

*(1) The model seems to apply different numerical schemes to different realms as inferred from Fig. 1. However, how the model does this is not 100% clear from the ms. Also, there seems to be some options that could affect the overall solution scheme/sequence, e.g., whether the model includes diffusion boundary layer or not; whether the advection is solved or not; whether the model tracks time or not, etc. Although the details are referred to the Supplement when relevant, a flow chart of the calculation scheme/sequence including branches for some important model options would be very helpful for the reader to grasp what is going on in the model.*

After rereading the text I agree that the description of how the different realms are connected to each other is incomplete in the text has to be improved. The different options do actually not always imply special branching, but generally lead to different codes to be generated, or to differ by small details only. The *volumeless* option, e.g., only uses a pre-compiler switch that adapts code at several places: it deactivates the evaluation and integration of the reaction rate term at the right-hand side of eq. (6) in one subroutine and switches between calculating the specific partial volumes of solids from their respective intrinsic densities, and setting them to zero in another. A flow chart would thus have to be very detailed, which would limit its usefulness, in my opinion. I think that these aspects can be more conveniently described in the text.

*(2) Pros and cons of options are discussed but it is not 100% obvious to the reader when to adopt which option. For example, in the ALL experiment, the author discussed the difference between the tracked time and actual time and the cause of it, but not sure when/why we should use the tracked time. Also, the author described that the implementation of mineral volume options (related to advection scheme) does not affect the overall results, which will make the reader*

*to wonder when/why to adopt which option regarding mineral volumes. Furthermore, under what conditions should we adopt the option of DBL? As a model description paper, providing a conclusion on the options may not be obligation, but guiding the reader a little bit more could be desired.*

Tracked time is rather expensive, as it adds a tracer to one solid and it requires all the processes that relate to that solid to be duplicated. If MEDUSA is only used as a ocean-sediment boundary scheme in a global biogeochemical cycle model, there is not stringent need to implement time tracking with CART in the sediment. However, if the produced sediment cores are meant to be compared to actual sediment data, tracked time would be recommended as it provides a reliable "age model" for the synthetic cores generated by the model. So, including or not including tracked time depends on the needs of the user.

The *volumeless* mineral option was only introduced to allow for a meaningful intercomparison with results derived from other models that do not take the effect of chemical reactions on the advection rate profile into account, but directly link the advection rate profile to the porosity profile via $w(z)(1 - \varphi(z)) = w_{\mathsf{SWI}}(1 - \varphi_{\mathsf{SWI}})$. Accordingly $w(z)$ is typically reduced by a factor of 2 to 2.5 only at depth, compared to $w_{\mathsf{SWI}}$. When the effect of chemical reactions is into account, that reduction can easily exceed a factor of 10. Volumeless solids are therefore not used by default. In the applications reported in the paper, the volumeless solids option is only used for the JEASIM, since the original model by Jourabchi et al. (2008) used such prescribed solids' advection rate profiles.

It is not entirely clear to me what is meant here by "does not affect the overall results". Selecting the volumeless option leads to significantly different advection rate profiles (easily different by a factor of five and more), and thus different concentration profiles (see lines 252*ff* and discussion in the

previous paragraph of this reply). If this comment refers to the statement at line 638 ("Both approaches are mathematically equivalent"), then this must be a misunderstanding. What is meant there is that prescribing a flux of inert material that matches the prescribed burial rate when the volumeless solids option leads to exactly the same advection rate profile that would be derived from the $w(z)(1 - \varphi(z)) = w_{\mathsf{SWI}}(1 - \varphi_{\mathsf{SWI}})$ identity.

The revised manuscript will include a new section discussing the role of a DBL (see also reply to comments by Anonymous Referee #1). That section will also provide guidance regarding other options offered by MEDUSA, and the volumeless option in particular.

**Specific comments**

*Table 1: In my understanding CaCO$_3$ and clay are not vertically resolved in Archer's original model (1991); only porewater chemistry and OM are vertically resolved. How do you define the layer number when layer numbers are different between different species as is the case for Archer (1991)?*

This is indeed correct–thank you for pointing out this imprecision. The number of grid points/layers is always taken as reported in the respective papers. In this particular case, that number relates thus only to solutes and organic carbon. Organic carbon distribution is furthermore calculated from the model of Emerson (1985), and used as a forcing function. Similar simplifications were made in the follow-up version (Archer, 1996): here again, calcite and detrital material were assumed to be homogeneous in the sedimentary mixed-layer, but this time the organic carbon profile was determined interactively. The relevant information will be amended in the Tables.

*Line 248: Are you saying that one of solid species is treated as a dilatant material and does not necessarily complies with advection law (e.g., Eq. (1) + equation in line 205)? I thought iterative implementation of Eqs. (1) and (6) and equation in line 205 could satisfy Eq. (4) (e.g., Archer et al., 2002).*

At first sight, one might indeed think that the solid species whose evolution equation is replaced by that for static volume conservation would possibly not comply to its original evolution equation. This is, however, fortunately not the case. The complete equation system at each grid point, which includes the evolution equations for the concentrations of all the solids and solutes under consideration (one instance of eq. (1) for each constituent), the equation for the advection rate at that grid point (eq. (6)) and the static volume equation (eq. (4)) is actually overdetermined. There is one more equation than there are unknowns (the constituents' concentrations and the advection rate). The equations are, however, not independent of each other: eq. (6) is obtained from the weighted average of the solids' evolution equations, weighted by the partial specific volumes of the respective solids, and taking the static volume equation into account, followed by vertical integration. Details about the relationships between these different equations are provided in the technical report '*Early Diagenesis in Sediments. A one-dimensional model formulation*' in the Supplement. One of the equations is thus redundant. We have chosen to strictly enforce the static volume conservation throughout the iterations and thus to keep its equation and drop one of the solids' evolution equations – the equation for the main inert material.

The procedure from Archer et al. (2002) is not guaranteed to always work out in a general purpose model. As reported by Archer et al. (2002) the actual total solid phase concentrations in MUDS could deviate by as much as $\pm 50\%$ from the actually required 1 g/g during the iterations. In a general

purpose approach as the one adopted in MEDUSA, where the complete process network is solved at once, such large deviations may lead to failure of the iterative process without any guardrails (here the static volume conservation equation). Furthermore, conserving mass to within 2% only may be sufficient for steady-state calculations, but could cause considerable model drift in transient simulation experiments. In MEDUSA, mass is typically conserved to within $10^{-12}$ and better for each constituent in each column and to within $10^{-10}$ or better globally for set-ups with several thousands of columns typical for 3D biogeochemical cycle models.

*Lines 258-270: Not sure whether the equations for different realms are solved at once or in a sequence with/without iterations. Is it appropriate to define a boundary condition for TRANSLAY as done for the above layers?*

All equations in the REACLAY realm (or in the combined REACLAY-DBL realms if a DBL is included) are solved at once. TRANLAY is indeed only a buffer reservoir that collects material leaving REACLAY through its bottom, or that feeds REACLAY. The equations that describe the evolution of its contents are thus ordinary differential equations (mass balance equations) that do indeed not require a boundary condition, but only source-minus-sink terms. This will be corrected in the text.

*Line 327: Do you mean $w$ is calculated time-explicitly but solids-solutes are calculated time-implicitly? Also, I suppose Newton iteration is conducted only in REACLAY and DBL? Do you separate calculations between REACLAY and DBL or at once?*

No, just like all the concentrations profiles, the $w$ **profile** is calculated time-implicitly.

Yes, **Newton iterations** are only used in the REACLAY and the DBL realms. The $w_k$ ($k$ denoting the grid vertices) could be calculated together with all the concentration profiles. During the early development stages of MEDUSA, the advection rates $w_k$ were treated on par with the concentrations of the model sediment components. The Newton-Raphson scheme was then based upon a Jacobian that also included derivatives of the equations with respect to the $w_k$'s. Please notice though that it was not eq. (6) that was used at that time to calculate the solids' advection rate profile, but its derivative (eq. (2.49) in the technical report *Early Diagenesis in Sediments – A one-dimensional model formulation* in the Supplement) which depends only on the local concentrations and not all of those above. The resulting discrete equation system too often became singular in the course of the iterations, preventing convergence. This is why we switched to eq. (6), which directly provides $w(z)$, but has the disadvantage of being dependent not only on local concentrations, but on all the concentrations above the level where $w(z)$ is calculated. The Jacobian of the equation system would thus be lower triangular with in addition some super-diagonal blocks, leading to Newton-Raphson steps that are computationally speaking orders of magnitude more expensive than the solution of a block tri-diagonal system. The iterative solution procedure for the system of equations was therefore split up: each iteration starts by updating the advection rate profile given the currently best available concentration profiles (a fixed-point approach for the $w_k$) and then a Newton-Raphson step is taken for the concentration values with that advection rate profile taken as given. For the next iteration, the advection rates are then again first updated using the new concentration profiles, followed by a Newton-Raphson step for the concentrations, etc.

In the revised manuscript, the contents of sections 2.1.1, 2.1.2 and 2.2 will be reorganised and section 2.2 partially rewritten to improve the description (also in reply to the comments by Anonymous Referee #1).

*Line 499: Is MEDUSA fully coupled to BEC (allowing exchange fluxes passed between the two models/modules) or are you just using the BEC output as boundary conditions and not returning any fluxes to BEC? The steady-state results of sedimentary profiles between the above two cases could different.*

Here, we are just using `BEC` output as boundary conditions for `MEDUSA`, as explained on lines 503–509: the coupling simulator only reads in the data from a file that would otherwise (i.e., in an actually coupled setup) provided by the host biogeochemical model. So there are no return fluxes to `BEC` (please notice that the `BEC` output used for the forcing dates from 2005).

Results from a fully coupled `BEC-MEDUSA` setup, albeit with a more recent version of `BEC` and further including $^{13}$C and $^{14}$C isotopes, with bi-directional exchange fluxes between the two models have been presented elsewhere (Kurahashi-Nakamura et al., 2020). As discussed at lines 567–570 in the manuscript, the results between the two cases are indeed strongly different and the two-way coupling reduces if not solves some of the shortcomings diagnosed here.

For the `COUPSIM` application, I actually consider that the model code is the central contribution to the paper as it illustrates how to couple `MEDUSA` to a biogeochemical model.

*Table 3, cap. L1: Isn't 1 L = 1 dm$^3$ universally correct? If so this does not have to be assumed?*

It actually *is* ... and has been so since 1964 already[1] ... So, there is indeed absolutely no need to assume this. That notice will be deleted in the revised manuscript.
* * *
[1]See https://www.bipm.org/en/CGPM/db/12/6/.

**Technical comments**

*Tables 1 and 2: 'BRNS-global' or 'BRNS-GLOBAL'?*

'BRNS-global' appears to be the correct spelling (Jourabchi, 2007). Corrected throughout.

I gladly accept all the other suggestions and corrections and will include them in the revised text.

**References**

Archer, D.: A data-driven model of the global calcite lysocline, Global Biogeochem. Cy., 10, 511–526, https://doi.org/10.1029/96GB01521, 1996.

Archer, D., Morford, J. L., and Emerson, S. R.: A model of suboxic sedimentary diagenesis suitable for automatic tuning and gridded domains, Global Biogeochem. Cy., 16, 1017, https://doi.org/10.1029/2000GB001288, 2002.

Archer, D. E.: Modeling the Calcite Lysocline, J. Geophys. Res., 96, 17 037–17 050, https://doi.org/10.1029/91JC01812, 1991.

Emerson, S.: Organic carbon preservation in marine sediments, in: The Carbon Cycle and Atmospheric $CO_2$ : Natural Variations Archean to Present, edited by Sundquist, E. T. and Broecker, W. S., vol. 32 of *Geophys. Monogr. Ser.*, pp. 78–87, AGU, Washington, DC, https://doi.org/10.1029/GM032p0078, 1985.

Jourabchi, P.: New Developments in Early Diagenetic Modeling : pH Distributions, Calcite Dissolution and Compaction, Ph.D. thesis, Utrecht University, Utrecht (NL), http://dspace.library.uu.nl/handle/1874/23398, 2007.

Jourabchi, P., Meile, C., Pasion, L. R., and Van Cappellen, P.: Quantitative interpretation of pore water $O_2$ and pH distributions in deep-sea sediments, Geochim. Cosmochim. Ac., 72, 1350–1364, https://doi.org/10.1016/j.gca.2007.12.012, 2008.

Kurahashi-Nakamura, T., Paul, A., Munhoven, G., Merkel, U., and Schulz, M.: Coupling of a sediment diagenesis model (MEDUSA) and an Earth system model (CESM1.2): a contribution toward enhanced marine biogeochemical modelling and long-term climate simulations, Geosci. Model Dev., 13, 825–840, https://doi.org/10.5194/gmd-13-825-2020, 2020.

---

## Author Response (AR1)

**Model of Early Diagenesis in the Upper Sediment with Adaptable complexity – MEDUSA (v. 2) : a time-dependent biogeochemical sediment module for Earth System Models, process analysis and teaching"**

**Author's Response**

Guy Munhoven

20th April 2021

Dear Sandra,

please find below my point to point listing of the changes made to the manuscript in response to the referees' comments and suggestions. For the sake of brevity, I do not repeat here the justifications that were given in the Author's Comments in reply to the Referees Comments. A *latexdiff* version of the manuscript highlighting the insertions and deletions in the text has also been uploaded alongside the revised manuscript.

I hope the manuscript is now acceptable for publication.

Best regards,

Guy

**Revisions in response to comments by Anonymous Referee #1**

**General Comments**

*2.2.2 Solution strategy*
*The different initial conditions for the iteration scheme are excellent and clear. However, I think it would be good if the convergence criteria is made more clear in the main manuscript. From your "Technical Reference" I understand that you test for a convergence of the (solid) concentration profiles. The second criterion, however, is difficult to comprehend. I think it would be good to include a clear description of both criteria in the main manuscript. Also, is the overall solution divided into different steps? E.g. is OM degradation and the resulting profile calculated first and from it MEDUSA deduces the O2 profile? Because the zonation of oxic and anoxic OM degradation has implications for the production of alkalinity and thus carbonate dissolution. Finally, does MEDUSA check for the convergence of burial velocity at the very end if the solid components are not declared as volumeless?*

- Reorganised the contents of sections 2.1 and 2.2 and furthermore rewrote large parts of section 2.2 to improve the description of the solution strategy and the two-level convergence criterion.
- Also extended chapter 3 of the "MEDUSA *Technical Reference*" to include explicit details about the scaling of solutes' equations.

**Specific/technical comments**

*ln. 21: I think it should read "the latter" here*

OK, corrected.

*ln. 46: Please give example reference for the meta-model approach.*

Added references to Sigman et al. (1998) and Dunne et al. (2007) (as announced in the AC1) and furthermore to Ridgwell (2007) and Capet et al. (2016).

*ln. 49 - 50: Please add references for examples for the "expert-chosen empirical parametric functions" and for the "system identification theory" approach*

- "expert-chosen empirical parametric functions": added reference to Dunne et al. (2007);
- "system identification theory": added references to Crucifix (2012) and to Ermakov et al. (2013) and rephrased sentence.

*Table 1+2 and in the corresponding text (e.g. ln 70 and first paragraph on pg. 5): For completeness please include OMEN-SED (Hülse et al., (2018) which is an analytical early diagenetic steady-state model for OM degradation dynamics with flexible resolution. OMEN-SED is available as stand-alone model and coupled to cGENIE and presents a noval approach to simulate benthic-pelagic coupling (i.e. different to the approaches presented in Table 1 and 2).*

- Added OMEN-SED in Tables 1 and 2.
- Added information about the coupling of OMEN-SED to cGENIE at lines 119–126 in the *latexdiff* report.

*Ln 127: "... during which some parts gets remineralised (i.e., oxidised or dissolved), and some parts gets buried." Please check the spelling here.*

Text modified to read "... during which some parts get remineralised (i.e., oxidised or dissolved), and the rest gets buried."

*Ln 130-133: Please add reference.*

Added reference to Archer et al. (1998).

*ln 134: "Previously buried carbonates will then return to the sedimentary mixed as a result of the bioturbation activity..." I guess the "sedimentary mixing" is a typo, please rephrase.*

"the sedimentary mixed" modified to read "the sedimentary mixed layer".

*Page 7: ln 142: I think the "instead of" can be deleted*

OK, deleted.

*Page 9: ln 197: So for solutes DBL is the same as REACLAY only that porosity equals 1? Could you please include a brief explanation why/when a DBL is important and when it should be used?*

Added new section 2.5 "Code building and customisation options: taming the flexibility" which also discusses the role and importance of a DBL.

*Page 10: ln 215: Pointing the reader here to you supplementary document "Early Diagenesis in Sediments "A one-dimensional model formulation" would be good as it provides more and very useful information for instance about the parameterisations for tortuosity and how the diffusion coefficients are calculated. Maybe also cite Ullman and Aller (1982) who did a lot of early work on diffusion coefficients and tortuosity parameterization.*

Rewrote text at lines 215*ff* (in the submitted manuscript, lines 268*ff* in the *latexdiff* report) and added footnote inviting to refer to the supplementary document for further details and references. Also included reference to Ullman and Aller (1982) to that document.

*ln. 221: bioirrigation "constant". What are you using for alpha in the set-up used in your simulations for this manuscript? Is it a depth dependent approach (e.g. Soetaert et al. (1996))?*

- Nothing changed around old l. 221, as the required information was already present.
- Added notice in the introductory paragraph of section 3 to precise that $\alpha$ is set to zero for the test case applications presented.

*Page 11: Additional constraints: Maybe it makes sense to include the constraint that the porosity profile is prescribed (i.e. time independent, $d\varphi/dt = 0$) at this point?*

Added short notice about the steady-state assumption on $\varphi$ at the end of the paragraph that presents the equation for determining the advection rate profile (now eq. (3) in section 2.1.1, heading "Transport") and non-steady state $\varphi$ would modify that equation.

*Page 12: ln. 292: "...and the topmost interior node of REACLAY" Is this part of the sentence not redundant as this note is always below the top of REACLAY? The same applies for the bottom of DBL.*

Nothing changed as I prefer the current text (see reply in AC1).

*Ln 297: "but the spacing and extent of each of these may be different." Depending on what? E.g. if the user wants to have a higher resolution for shelf-sediments compared to the deep ocean?*

No change required.

*Page 16: ln 405: I think this should read: dissolved organic matter instead of "dissolved inorganic matter"*

OK, corrected as suggested.

*Page 17: ln 438: I think it should read: simulation experiments from Munhoven (2007)*

Suggestion incorrect as explained in the AC1 – nothing changed.

*ln 450-451: I guess the sentence could be changed to: "MBM is an eleven-box carbon cycle model of the carbon cycle in the ocean and the atmosphere"*

Text changed to "MBM is an eleven-box model of the carbon cycle in the ocean and the atmosphere"

*ln 453: close the parenthesis after Pacific*

OK, corrected.

*Section 3.2 COUPSIM — Coupling simulator page 21 ln. 519: why did you just simulate depths greater than 1000m below sea-level? And are the shallower observations in Seiter (where generally the highest TOC concentrations are observed) excluded from the data-set (or is there hardly any data for shallower settings because of the resolution of the observations)?*

Added explanatory sentence regarding this choice (lines 700*ff* in the *latexdiff* document).

*Fig. 3 : I think opal are the red triangles. In that case the caption needs to be corrected.*

OK, corrected.

*Ln 551: Your rate constants for OM degradation could be for instance compared to Palastanga et al. (2011) who used the HAMOCC model coupled to a diagenetic model using a 1G-approach for OM. They also compare their results to the Seiter et al. (2005) data and find the best fit using k_ox=0.005 yr-1 and k_anox=0.002 yr-1 for depths > 2000m (k_ox=0.01 yr-1 and k_anox=0.008 yr-1 for depth < 2000m).*

Amended paragraph (see lines 735*ff* in the *latexdiff* report) to include that comparison.

*page 27 ln 652: spelling: I think it should read: "... in order to allow..."*

OK, corrected.

**Revisions in response to comments by Anonymous Referee #2**

**General comments**

*(1) The model seems to apply different numerical schemes to different realms as inferred from Fig. 1. However, how the model does this is not 100% clear from the ms. Also, there seems to be some options that could affect the overall solution scheme/sequence, e.g., whether the model includes diffusion boundary layer or not; whether the advection is solved or not; whether the model tracks time or not, etc. Although the details are referred to the Supplement when relevant, a flow chart of the calculation scheme/sequence including branches for some important model options would be very helpful for the reader to grasp what is going on in the model.*

> Reorganised the contents of sections 2.1 and 2.2 and rewrote large parts of section 2.2 to provide more details about the solution strategy. No flow chart added, as I am not convinced about the usefulness of such a scheme for a complex model such as MEDUSA.

*(2) Pros and cons of options are discussed but it is not 100% obvious to the reader when to adopt which option. For example, in the ALL experiment, the author discussed the difference between the tracked time and actual time and the cause of it, but not sure when/why we should use the tracked time. Also, the author described that the implementation of mineral volume options (related to advection scheme) does not affect the overall results, which will make the reader to wonder when/why to adopt which option regarding mineral volumes. Furthermore, under what conditions should we adopt the option of DBL? As a model description paper, providing a conclusion on the options may not be obligation, but guiding the reader a little bit more could be desired.*

> Added a new section 2.5 ("Code building and customisation options: taming the flexibility") to offer some guidance to potential users about the most common options, including a discussion on the importance of a DBL

**Specific comments**

*Table 1: In my understanding $CaCO_3$ and clay are not vertically resolved in Archer's original model (1991); only porewater chemistry and OM are vertically resolved. How do you define the layer number when layer numbers are different between different species as is the case for Archer (1991)?*

> - Added notice to the caption of Table 1 regarding the meaning of the number of layers/nodes.
> - Amended information about the model composition in Archer (1991) in Table 2. Similarly corrected information about Archer (1996).

*Line 248: Are you saying that one of solid species is treated as a dilatant material and does not necessarily complies with advection law (e.g., Eq. (1) + equation in line 205)? I thought iterative implementation of Eqs. (1) and (6) and equation in line 205 could satisfy Eq. (4) (e.g., Archer et al., 2002).*

> Added explanatory paragraph in section 2.2.1 (at the end of "Discrete equations", ll. 377–392 in the *latexdiff* report below) about the over-determination of the equation system, making one of the equations redundant.

*Lines 258-270: Not sure whether the equations for different realms are solved at once or in a sequence with/without iterations. Is it appropriate to define a boundary condition for TRANSLAY as done for the above layers?*

Section 2.2 largely rewritten to make it clearer how the equation systems for the different realms are solved, and how their solutions influence each other.

*Line 327: Do you mean $w$ is calculated time-explicitly but solids-solutes are calculated time-implicitly? Also, I suppose Newton iteration is conducted only in REACLAY and DBL? Do you separate calculations between REACLAY and DBL or at once?*

Solution strategy described in more detail in the reorganised and extensively rewritten section 2.2

*Line 499: Is MEDUSA fully coupled to BEC (allowing exchange fluxes passed between the two models/modules) or are you just using the BEC output as boundary conditions and not returning any fluxes to BEC? The steady-state results of sedimentary profiles between the above two cases could different.*

No changes made, as this information was already included in the text.

*Table 3, cap. L1: Isn't $1\,L = 1\,dm^3$ universally correct? If so this does not have to be assumed?*

Deleted sentence regarding this "assumption" in the caption to Table 3.

**Technical comments**

*Line 134: 'the sedimentary mixed' should be replaced with 'the sedimentary mixed layer'?*

OK, corrected.

*Line 453: Right parenthesis is missing.*

OK, corrected.

*Tables 1 and 2: 'BRNS-global' or 'BRNS-GLOBAL'?*

Spelling changed to 'BRNS-global' throughout.

*Fig. 3 caption L2: %Opal should be red triangles?*

Yes, they should – corrected.

**Author's own changes**

Line 16 (also line 16 in the *latexdiff* report):

"Ocean-atmosphere exchange schemes" corrected to read "Ocean-sediment exchange schemes"

Table 2:

Added missing $CO_2$ and $HCO_3^-$ to the list of solutes in MUDS.

Table 2:

Added notice to the caption stating that "Clay" stands for inert, detrital or dilutant material.

Line 259 (line 306 in the *latexdiff* report):

"ovelying" corrected to read "overlying"

Line 263 (line 310 in the *latexdiff* report):

"accross" corrected to read "across"

Line 270 (line 318 in the *latexdiff* report):

"$wz_B^+$" corrected to read "$w_B^+$"

Line 596 (line 781 in the *latexdiff* report):

"oxygen concentration rates" corrected to read "oxygen concentrations"

Throughout:

Changed "seafloor" to read "sea-floor"

*Code availability* section:

Archived the codes in three archives on Zenodo and amended the section text accordingly, giving the DOIs of the three archives, that had to be uploaded separately because of the different licenses.

Supplement:

- Added reference manual for the $\mu$XML library in `muxml/doc`.
- Updated `buildandrun.pdf` to include information on getting and assembling the codes from three Zenodo code archives in order to run the test case applications.
- Cosmetic changes in `medusa-cocogen.pdf` to fix overfull lines.
- Code archive `medusa_v2.tar.gz`: corrected a glitch in `apps/jeasim/Makefile` that prevented the JEASIM application from compiling.
- Updated `smcontents.pdf` to reflect the above changes; also added versions and/or dates of the different documents.

**References**

Archer, D.: A data-driven model of the global calcite lysocline, Global Biogeochem. Cy., 10, 511–526, https://doi.org/10.1029/96GB01521, 1996.

Archer, D., Kheshgi, H., and Maier-Reimer, E.: Dynamics of Fossil Fuel $CO_2$ Neutralization by Marine $CaCO_3$, Global Biogeochem. Cy., 12, 259–276, https://doi.org/10.1029/98GB00744, 1998.

Archer, D., Morford, J. L., and Emerson, S. R.: A model of suboxic sedimentary diagenesis suitable for automatic tuning and gridded domains, Global Biogeochem. Cy., 16, 1017, https://doi.org/10.1029/2000GB001288, 2002.

Archer, D. E.: Modeling the Calcite Lysocline, J. Geophys. Res., 96, 17 037–17 050, https://doi.org/10.1029/91JC01812, 1991.

Capet, A., Meysman, F. J. R., Akoumianaki, I., Soetaert, K., and Grégoire, M.: Integrating sediment biogeochemistry into 3D oceanic models : A study of benthic-pelagic coupling in the Black Sea, Ocean Model., 101, 83–100, https://doi.org/10.1016/j.ocemod.2016.03.006, 2016.

Crucifix, M.: Traditional and novel approaches to palaeoclimate modelling, Quaternary Sci. Rev., 57, 1–16, https://doi.org/10.1016/j.quascirev.2012.09.010, 2012.

Dunne, J. P., Sarmiento, and Gnanadesikan, A.: A synthesis of global particle export from the surface ocean and cycling through the ocean interior and on the seafloor, Global Biogeochem. Cy., 21, GB4006, https://doi.org/10.1029/2006GB002907, 2007.

Ermakov, I., Crucifix, M., and Munhoven, G.: Emulation of the MBM-MEDUSA model: exploring the sea level and the basin-to-shelf transfer influence on the system dynamics, Geophysical Research Abstracts, EGU2013-9011-2, URL https://meetingorganizer.copernicus.org/EGU2013/EGU2013-9011-2.pdf, 2013.

Palastanga, V., Slomp, C. P., and Heinze, C.: Long-term controls on ocean phosphorus and oxygen in a global biogeochemical model, Global Biogeochem. Cy., 25, GB3024, https://doi.org/10.1029/2010GB003827, 2011.

Ridgwell, A.: Interpreting transient carbonate compensation depth changes by marine sediment core modeling, Paleoceanography, 22, PA4102, https://doi.org/10.1029/2006PA001372, 2007.

Sigman, D. M., McCorkle, D. C., and Martin, W. R.: The calcite lysocline as a constraint on glacial/interglacial low-latitude production changes, Global Biogeochem. Cy., 12, 409–427, https://doi.org/10.1029/98GB01184, 1998.

Ullman, W. J. and Aller, R. C.: Diffusion coefficients in nearshore marine sediments1, Limnol. Oceanogr., 27, 552–556, https://doi.org/10.4319/lo.1982.27.3.0552, 1982.